# FusionShot: Boosting Few Shot Learners with Focal-Diversity Optimized Ensemble Method

## Abstract

Designing optimal few-shot learners is challenging. First, it is hard to train a few-shot model that can deliver the best generalization performance on all benchmarks compared to existing state-of-the-art (SOTA) methods. Second, unlike traditional deep neural networks (e.g., CNN, auto-encoder), few-shot learners utilize the metric space distance-based loss function to optimize the deep embedding learning on complex or multi-modal data. Both the choice of latent similarity computation methods and the choice of DNN embedding algorithms for latent feature extraction will impact the generalization performance of few-shot learners. This paper presents FusionShot, a focal diversity optimized few-shot ensemble learning framework with three original contributions. First, we revisit the few-shot learning architectures to analyze why some few-shot learners perform well whereas other SOTA few-shot models fail miserably. Second, we explore and compare two alternative fusion channels to ensemble multiple few-shot learners: (i) the fusion of various latent distance methods, and (ii) the fusion of multiple DNN embedding algorithms that learn/extract latent features differently. Finally, we introduce a focal-diversity optimized few-shot ensemble learning framework for further boosting the performance of few-shot ensemble learning. Extensive experiments on representative few-shot benchmarks (mini-Imagenet and CUB) show that our FusionShot can select the best performing ensembles from a pool of base few-shot models, which outperform the representative SOTA models, on novel tasks (unknown at training), even when a majority of the base models fails. For reproducibility purposes, trained models, results, and code are made available at `https://anonymous.4open.science/r/fusionshot-0A44/`.

## 1 Introduction

Few-shot learning is gaining increased attraction for its remarkable ability to perform novel tasks that are unknown during model training, including recent advances in large language models, e.g., ChatGPT4 (OpenAI, 2023). However, in image classification, designing an optimal few-shot learner that can persistently outperform the state-of-the-art (SOTA) methods seems stalled in recent years for a number of reasons. First, it is extremely hard to train an individual few-shot learner that can compete and win persistently. Second, unlike conventional deep neural networks (DNNs), e.g., auto encoder-decoder and CNN, which learn invariant and distinctive features about data by building high-level features from low-level ones through learning hierarchical feature representations; the few-shot learner employs the metric space learning, a distance metric based approach, to learn deep embedding of input data for latent feature extraction. By combining deep embedding learning with metric space distance learning through metric-space loss optimization, the few-shot learners can map examples of similar features in the real world (input data space) to the latent neural embedding representations with dual properties: (i) the examples of similar features in the real world will be mapped to the latent feature vectors that are closer in the latent embedding space, and (ii) the examples with large dissimilarity in the real world will be sharply distant in the latent feature embedding space. This empowered the few-shot learners to classify samples with limited supervision on novel tasks that are completely unknown during model training.

**Related Work.** The most relevant few shot algorithms include the recent SOTA individual few-shot algorithms, e.g., SimpleShot (Wang et al., 2019), DeepEMD (Zhang et al., 2020), and optimizations (Bateni et al., 2020; Ye et al., 2020), and the recent few-shot ensemble methods, e.g., Robust-20 (Dvornik et al., 2019). They improve the top-1 or top-5 performance over the traditional few-shot learners, e.g., ProtoNet (Snell et al., 2017), MatchingNet (Vinyals et al., 2016), and re-

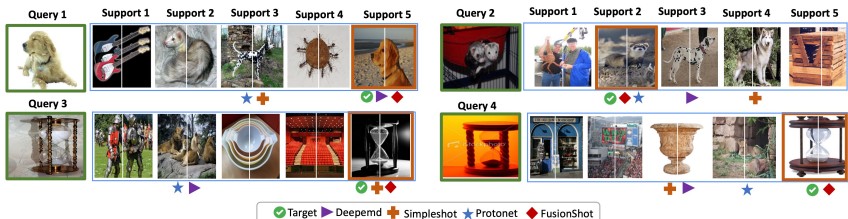

Figure 1: Understanding Individual Few-shot models with illustrative Examples

lationNet (Sung et al., 2018). The key distinguishing property of these recent improvements can be characterized from three observations. First, recent studies suggest to use more complex latent distance function, such as Earth Mover's Distance (Zhang et al., 2020), Mahalanibis (Bateni et al., 2020), and other methods (Ye et al., 2020) for learning latent feature similarity. For example, DeepEMD (Zhang et al., 2020) shows that under the same backbone DNN feature extractor, Earth Mover's Distance (EMD) metric performs better than other non-parametric distance metrics, e.g., cosine, Euclidean, and some popular parametric distance metrics, e.g., the relationNet (Sung et al., 2018). Second, SimpleNet (Wang et al., 2019) shows that using k-nearest-neighbor and k-means clustering with Cosine distance can be more effective for learning metric-space distance than traditional few-shot algorithms, such as MatchingNet, ProtoNet, RelationNet. Finally, other studies also suggest leveraging supervised pre-training via transfer learning to optimize the initialization of the DNN embedding backbone architecture for latent feature extraction and latent feature similarity learning (Chen et al., 2019). Interestingly, most of the SOTA models fail to offer consistent and stable performance. Figure 1 compares four few-shot models: ProtoNet, DeepEMD, SimpleShot, and FUSIONSHOT (proposed in this paper). For Query-1, DeepEMD and FUSIONSHOT succeed in top-1 prediction while SimpleShot and ProtoNet both fail. However, DeepEMD fails on Query-2, whereas all others succeed with the correct top-1 result, and Query-3, whereas ProtoNet also fails, but SimpleShot and FUSIONSHOT succeed. Query-4 is a novel test example where DeepEMD, SimpleShot, and ProtoNet all fail, but FUSIONSHOT succeeds in finding the correct top-1 result by integrating focal diversity optimized ensemble pruning and learn to combine.

**Scope and Contributions.** This paper propose FUSIONSHOT, a focal diversity optimized few-shot ensemble learning framework and algorithms, by leveraging complimentary wisdom of multiple few-shot models through two alternative channels of fusion methods. We aim to create the fusion of feature learning embeddings by using either (i) the same feature extraction DNN architecture with different latent distance functions or (ii) the different feature extraction architectures combined with the same distance function. By revisiting the few-shot learning architectures and understanding when and why the representative few-shot models may fail, we propose a learn-to-combine methodology for resolving the prediction disparity among multiple diverse few-shot models, and generate robust prediction through our ensemble fusion on novel datasets that are unknown at training phase. We further boost our ensemble fusion performance by introducing a focal-diversity optimized few-shot ensemble pruning method. The proposed focal diversity metrics can accurately capture the negative correlation among component models of a few-shot ensemble and ensure with high confidence that an ensemble of high focal diversity will have high error diversity, low negative correlation, and strong failure independence. To the best of our knowledge, Existing few-shot ensemble methods are mainly based on joint training of multiple feature extractors (Dvornik et al., 2019; Bendou et al., 2022). FUSIONSHOT is the first to ensemble multiple independently trained few-shot models through integrating learning to combine and focal diversity optimization. We evaluate the effectiveness of FUSIONSHOT by comparing it with both existing individual SOTA few-shot models and the representative few-shot ensemble methods on popular benchmarks (mini-Imagenet and CUB). Our results show that given a pool of base few-shot models, our FUSIONSHOT can select the best few-shot sub-ensembles, which offer better generalization performance on novel tasks, unknown at training, compared to the existing SOTA few-shot learners and few-shot ensemble models, even when the majority of the base models in a FUSIONSHOT ensemble team fails.

## 2 FEW-SHORT LEARNING: THE REFERENCE ARCHITECTURE

Few-shot algorithms learn to extract latent features of complex data through joint training of a DNN embedding function $f_\theta$ and a latent feature distance function $g_\psi$. The embedding function should preserve the distance of input data in their latent embedding space such that (i) the pair of input data should be close in the latent feature space and (ii) the input data with high similarity defined by the

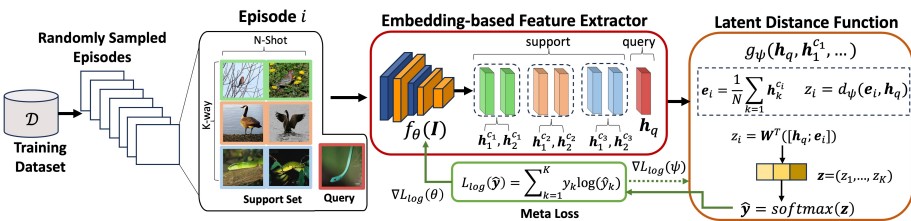

Figure 2: A Reference Framework for Few Shot Learning

pairing relation should be also similar in the latent embedding space. Figure 2 gives a sketch of the few-shot learning reference architecture. For classification, the few-shot learner is optimized to keep the pairwise distance between samples from the same class much closer in the latent embedding space than the pairwise distance between samples of different classes. This empowers the trained model capable of transferring knowledge to novel data of UNSEEN classes.

We can formulate the problem as follows: Let $\mathcal{D} = \{(\mathbf{I}_i, y_i)\}_{i=1}^{L}$ denote a dataset with each sample $\mathbf{I}_i$ paired with a corresponding class label $y_i$ and $y_i \in C = \{c_1, c_2, \ldots, c_L\}$. Note that instead of pairing with a label, one can also pair with another input for self-supervised learning, such as a pair of food image and recipe text (Xie et al., 2021). To train a few shot learner, we first partition $\mathcal{D}$ into $\mathcal{D}^{\mathrm{train}}$, $\mathcal{D}^{\mathrm{val}}$, and $\mathcal{D}^{\mathrm{novel}}$, denoting training set, validation set, and novel test set respectively, and $L^{train} + L^{val} + L^{novel} = L$. Let $B$ denote the number of samples per class. We have $|\mathcal{D}^{\mathrm{train}}| = L^{train} \times B$, $|\mathcal{D}^{\mathrm{val}}| = L^{val} \times B$, and $|\mathcal{D}^{\mathrm{novel}}| = L^{novel} \times B$. The few-shot learning consists of two stages: episode-based meta-learning and episode-based meta-testing (Vinyals et al., 2016). In meta-learning, we train the model by using $\mathcal{D}^{\mathrm{train}}$, and validate its performance on the $\mathcal{D}^{\mathrm{val}}$, including tuning hyper-parameters. In meta-testing, the performance of the trained few-shot model will be tested on $\mathcal{D}^{\mathrm{novel}}$. Figure 2 gives a sketch of the *episode* based few-shot learning architecture. We call one forward pass as an episode. For $K$-*way $N$-shot* learning ($1 \leq K << min\{L^{train}, L^{val}, L^{novel}\}, 0 \leq N << B$), an episode consists of a query $\mathbf{Q}$ and a support set $\mathcal{S}$ of $K$ classes with $N$ samples per class. Both $\mathbf{Q}$ and $\mathbf{S}$ are randomly sampled from $\mathcal{D}^{\mathrm{train}}$ or $\mathcal{D}^{\mathrm{val}}$ or $\mathcal{D}^{\mathrm{novel}}$ to create the training episode set, the validation episode set, or the novel episode set for testing respectively. The composition of an episode should meet the following constraints: First, $\mathcal{S} = \{\{\mathbf{I}_1^{c_1}, \ldots, \mathbf{I}_N^{c_1}\}, \ldots, \{\mathbf{I}_1^{c_K}, \ldots, \mathbf{I}_N^{c_K}\}\}$, and $|\mathcal{S}| = N \times K$. Let $\mathbf{I}_i^{c_j}$ denote an input sample $i$ belonging to class $c_j$. After creating the support set $\mathcal{S}$, a query $\mathbf{Q}$ is randomly sampled to pair with $\mathcal{S}$ for composing the episode. The query $\mathbf{Q}$ should meet the following criteria: (i) $\mathbf{Q}$ should be drawn from samples in the training set, i.e., $\mathbf{Q} \in \mathcal{D}^{\mathrm{train}} \setminus \mathcal{S}$; and (ii) $\mathbf{Q}$ should refer to a sample that does not exist in the support set $\mathcal{S}$ but belongs to one of the $K$ classes used in $\mathcal{S}$, i.e., $\mathbf{Q} \notin \mathcal{S}, \mathbf{Q} = \mathbf{I}^{c_j} c_j \in \{c_1, \ldots, c_K\}, \mathcal{S} \subset \mathcal{D}^{\mathrm{train}}$. The number of episodes used per epoch for few-shot learning on both mini-Imagenet and CUB benchmark is $1,600$, $1,600$, and $600$ for training, validation, and novelty testing respectively. Episode-based training for a $K$-way $N$-shot learner will iterate through all training episodes. The total number of iterations is a hyper-parameter, usually larger than one epoch, and is set differently by different algorithms (detail in Appendix).

For each iteration, we send one episode $(\mathcal{S}, \mathbf{Q})$ to the DNN embedding function $f_\theta(.)$ to generate a total of $|\mathcal{S}| + 1$ latent embedding mappings, denoted by $\mathcal{H}_i = \{\mathbf{h}_1^{c_i}, \ldots, \mathbf{h}_N^{c_i}\}$, where $\mathbf{h} = f_\theta(\mathbf{I}^{c_i})$ $(i = 1, \ldots, K)$, plus one query embedding $\mathbf{h}_q = f_\theta(\mathbf{Q})$, as shown in the middle of Figure 2. Second, the latent distance function $g_\psi(.)$ takes all $N$ latent embeddings of the same class $c_i$ to obtain a per-class integrated latent embedding by taking the averaging, for each of the $K$ classes in the support set, denoted by $\{\mathbf{e}_{c_1}, \ldots \mathbf{e}_{c_i}, \ldots, \mathbf{e}_{c_K}\}$. Then we use a latent-distance-method $z = d_\psi(\mathbf{h}_q, \mathbf{e}_{c_i})$ to compute the distance between query embedding $\mathbf{h}_q$ and each of the $K$ class embedding $\mathbf{e}_{c_i}$. Followed by *softmax* and the entropy based loss, we obtain the meta-loss and perform stochastic gradient descent and back-propagation to initiate the next iteration of the learning with a new episode. This training process repeats until all training episodes are consumed. The latent-distance-method $d_\psi(.)$ is typically a neural network based on $L2$ or $Cosine$ dos (Sung et al., 2018; Vinyals et al., 2016; Koch et al., 2015; Snell et al., 2017). Thus, the DNN embedding function $f_\theta(.)$ with parameters $\theta$ and the latent similarity function $g_\psi$ with parameters $\psi$ are jointly trained.

## 3 DESIGN OVERVIEW OF FUSIONSHOT

In contrast to existing few-shot ensemble method that uses joint-model training (Dvornik et al., 2019), FUSIONSHOT presents a novel design for few-shot ensemble learning from three perspectives. (1) We use a selection of $M$ independently trained few-shot models as a pool of base models.

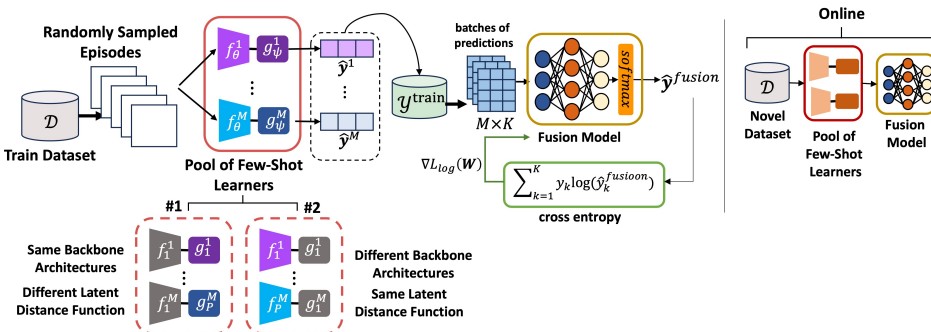

Figure 3: An overview of FUSIONSHOT framework for learning ensemble fusion.

(2) We further optimize our ensemble learning through focal-diversity optimized ensemble pruning. Instead of constructing an ensemble of all $M$ base models, we utilize a focal-diversity optimized method to identify those base models that often fail together through focal-diversity based ensemble pruning, which recommends only those sub-ensembles of high focal diversity. (3) Instead of using non-parametric consensus like the majority voting to generate ensemble prediction to each novel query based on the $m$ independent predictions from an ensemble team of size $m$, we use a parametric approach that learns to combine multiple independently trained models for different ensemble team size $m$ $(2 \le m \le M)$. In this section we describe the first and the third design choices and defer the discussion on the focal-diversity optimized ensemble pruning to Section 4. Figure 3 gives a sketch of FUSIONSHOT design.

The first prototype of FUSIONSHOT ensemble method is dedicated to explore two alternative fusion channels to construct a few-shot ensemble: the use of multiple latent distance functions, and the use of multiple latent feature extractors. Both fusion channels will use a parametric approach to building an ensemble fusion model, which learns to combine $m$ independently trained few-shot models and then generate the ensemble output for each novel episode.

**Few-shot ensemble by latent distance fusion.** Given a novel episode $i$, denoted by $(\mathbf{Q}_i, \mathcal{S}_i)$, a few-shot ensemble learner of $M$ component models will send the episode $i$ to all $M$ individual component models. Let each model $j$ produce the output $\hat{\mathbf{y}}_i^j = p(\mathbf{y}|\mathcal{S}_i, \mathbf{Q}_i; \theta)$, where $\hat{\mathbf{y}}_i^j \in \mathbb{R}^K$ is the probability vector. Each value in $\hat{\mathbf{y}}_i^j$ indicates the probability of the query $\mathbf{Q}_i$, belonging to a class present in the support set $\mathcal{S}_i$. For each episode $i$, there are $M$ component models and $M$ different predictions against the query $\mathbf{Q}_i$ over the support set $\mathcal{S}_i$. Thus, we have $\hat{\mathbf{y}}_i^1, \ldots, \hat{\mathbf{y}}_i^M$. The goal of the FUSIONSHOT ensemble model is to learn the most robust way to combine the $M$ different predictions to generate the ensemble output against the query $\mathbf{Q}_i$ for each episode in $E_{novel}$. Specifically, for an episode $i$, the objective is to maximize $p(y_i = k|\hat{\mathbf{y}}_i^1, \ldots, \hat{\mathbf{y}}_i^M)$ for mapping the query $\mathbf{Q}_i$ to the class $c_k$. Here, we parameterize the likelihood with a Multi-Layer Perception (MLP), $p(y_i = k|\hat{\mathbf{y}}_i^1, \ldots, \hat{\mathbf{y}}_i^M; \theta_{ens})$, to approximate the probability. We want to find the best parameters $\theta_{ens}$ to maximize the likelihood, which can be reduced to minimize the cross-entropy loss (see Equation 3), and to optimize the parameters of the ensemble model. This approach also enables FUSIONSHOT ensemble neural network to learn the novel ways to combine and to make correct prediction even when a majority of the $M$ component models fail.

**Few-shot ensemble by leveraging multiple feature extraction embedding methods.** Assume that we want to construct a few-shot ensemble from $P_{fe}$ different feature extraction embedding methods and $P_{ld}$ different latent distance methods, there are $P_{fe} \times P_{ld}$ number of component models which we can choose to create a pool of base models. For example, if we choose $M$ different DNN backbones to generate $M$ different feature extraction embedding for each episode $i$, then we can choose also $M$ different latent distance methods, one corresponding to each of the $M$ backbone algorithms. This will represent the most general way of constructing a few-shot ensemble, as shown in the top left of Figure 3. Alternatively, we can choose ResNet18 as the single backbone DNN for learning feature extraction embedding, and choose $M$ different latent distance methods, e.g., protoNet, relationNet, MatchingNet, SimpleNet, DeepEMD, each is jointly trained with ResNet18, and hence, we have also $M$ different few-shot models, all with the same DNN embedding method (ResNet18), but each has its own latent distance method. This will allow us to build a few-shot ensemble learner of $M$ few-shot models through the fusion of the $M$ latent distance methods, as illustrated in Figure 3 under the #1 scenario of the general ensemble method. Similarly, by using

$M$ different backbone algorithms to learn $M$ different feature extraction embedding, each is jointly learned with one latent distance method, say simpleShot, we can create the #2 scenario.

**Ensemble Fusion through Learning.** In FUSIONSHOT, we first determine the fusion channel to use for creating a pool of $M$ base models ($M$ is a configurable hyper-parameter). Then we will construct the FUSIONSHOT specific training dataset to train the FUSIONSHOT model for effective learning of few-shot ensemble fusion. We learn a few-shot ensemble model in four steps. (1) We choose $M$ independently trained few-shot models as the pool of $M$ base models. For example, the models can be trained using one or $M$ different backbone algorithms for feature extraction embedding and $M$ different latent distance methods, as shown in Figure 3. (2) We feed the training episodes, one at a time, to the $M$ component models. For each episode $i \in E_{train}$, denoted by $(\mathbf{Q}_i, \mathcal{S}_i)$, we collect the query prediction probability vector of size $K$, denoted by $(\hat{\mathbf{y}}_i^1, \dots, \hat{\mathbf{y}}_i^K)$, each of the $K$ values corresponds to the confidence of matching the query $\mathbf{Q}_i$ to one of the $K$ classes in the support set $\mathcal{S}_i$. Let $L_{train}$ denote the number of training episodes, and $M$ is the total number of base few shot models. We have the ensemble fusion training set $\mathcal{Y}^{train} = \{\hat{\mathbf{y}}_i^j\}_{i=1,j=1}^{M,L_{train}}$. (3) We feed the ensemble fusion training data collected for learning to combine using a MLP network with cross-entropy loss optimization, as illustrated in Figure 3. (4) We use the validation episodes and novel episodes to create $\mathcal{Y}^{val}$ and $\mathcal{Y}^{novel}$. We use the predictions of the novel dataset, $\mathcal{Y}^{novel}$, as the performance evaluation of each ensemble model in FUSIONSHOT.

## 4 FEW-SHOT ENSEMBLE PRUNING WITH FOCAL DIVERSITY

Given a pool of $M$ base models, the total number of ensemble teams with size $m$ ($2 \le m \le M$) is $2^M - M - 1$. The number of ensemble sets grows exponentially as $m$ gets larger, e.g., when $M$ grows from 3 to 6 to 10, the number of candidate ensemble teams will be 4, 57, 1013 respectively. Ensemble pruning is critical for several reasons. First, a recent study shows that the ensemble of $M$ base models may not outperform some sub-ensembles of size $m$ ($m << M$) (Wu et al., 2021). Second, ensemble pruning may effectively narrow down to the small selection of good ensemble teams. A key question is how to effectively perform ensemble pruning.

### 4.1 FOCAL NEGATIVE CORRELATION AND FOCAL DIVERSITY

In FUSIONSHOT we introduce two episode-based disagreement metrics: the focal negative correlation metric, $\sigma^{focal}$, and the focal diversity metric $\lambda^{focal}$. The former is used to quantify the level of error diversity among the component models of an ensemble with respect to each model within the ensemble. The latter is used to quantify the general error diversity of the ensemble by taking into account all focal negative correlation scores of an ensemble. Consider a few-shot ensemble $\mathcal{E}^m$, composed of $m$ models: $\{FS_1, \dots, FS_i, \dots, FS_m\}$, we choose one of the $m$ base models each time as the focal model to compute the focal negative correlation score of this ensemble, denoted as $\sigma^{focal}(\mathcal{E}_m; FS_i)$. We define the focal diversity of this ensemble team by the average of the $m$ focal negative correlation scores. The procedure of computing the focal negative correlation score of $\sigma^{focal}$ is as follows: (i) select a base model among the set of $m$ base models as the *focal* model, (ii) take all the validation episodes that the focal model has failed and calculate the focal negative correlation score, (3) repeat the previous steps until all $m$ focal negative correlation scores are obtained. $\{\sigma_1^{focal}, \dots, \sigma_m^{focal}\}$, and (4) compute the average over $\{\sigma_1^{focal}, \dots, \sigma_m^{focal}\}$ to obtain the focal diversity of ensemble $\mathcal{E}^m$, denoted by $\lambda^{focal}(\mathcal{E}^m)$:

$$\lambda^{focal}(\mathcal{E}^m) = 1/m \times \sum_{i=1}^{m} \sigma^{focal}(\mathcal{E}^m; FS_i) \tag{1}$$

$$\sigma^{focal}(\mathcal{E}^m; FS_i) = 1 - \sum_{j=1}^{M} \frac{j(j-1)}{m(m-1)} p_j \Big/ \sum_{j=1}^{M} \frac{j}{M} p_j$$

Here $p_i$ is the probability that $i$ number of models fail together on a randomly chosen episode. We calculate as $p_i = n_i/L^{val}$ where $n_i$ is the total number of episodes that $i$ number of models failed together on the $\mathcal{Y}^{val}$ and $L^{val}$ is the total number of validation episodes. The nominator in $\sigma^{focal}$ represents the probability of two randomly chosen models simultaneously failing on an episode, while the denominator represents one randomly chosen model failing on an episode. The terms beneath $p_j$ values are the probability of the chosen model being one of the failures. For example, when $M = 3$, there are three cases of model failures; one, two, or three models can fail simultaneously. If one model fails, the chance of selecting the failed model is $1/3$. Similarly, for two models, it is $2/3$, and for three models, it is $1$. In the case of minimum diversity, the probability of two randomly chosen models failing together comes down to the probability of one of them failing,

which makes the fraction term equal to 1 and $\sigma^{focal} = 0$. Similarly, in the case of maximum diversity, there are no simultaneous failures. Hence, the nominator equals 0 and $\sigma^{focal} = 1$.

## 4.2 ENSEMBLE PRUNING STRATEGY AND OPTIMIZATION

Our goal is not only to select the ensemble with its component models making the most diverse wrong predictions but also to encourage cooperation by working together to produce correct answers. For example, an ensemble of two models can be diverse but failing when the two base models make wrong predictions even though they may not have the same wrong prediction. In this scenario, strategies that advocate cooperation is essential to capture and promote the collaborative behavior among the base models for boosting ensemble fusion performance. To this end, we enhance the ensemble of component models by incorporating the plurality voting into our focal diversity optimized ensemble pruning method. The plurality voting selects the most voted decision to recommend as the final decision. If there is no plurality, then it randomly picks among the decisions of the $m$ component models of the ensemble $\mathcal{E}^m$. Also, we can use plurality voting to examine the validation episodes to create a lower bound for the selection of the best ensemble models, which allows us to filter the worst ensemble sets rapidly before running focal diversity ensemble pruning. Figure 4a shows the plurality voting accuracy against focal diversity values of all candidate ensem-

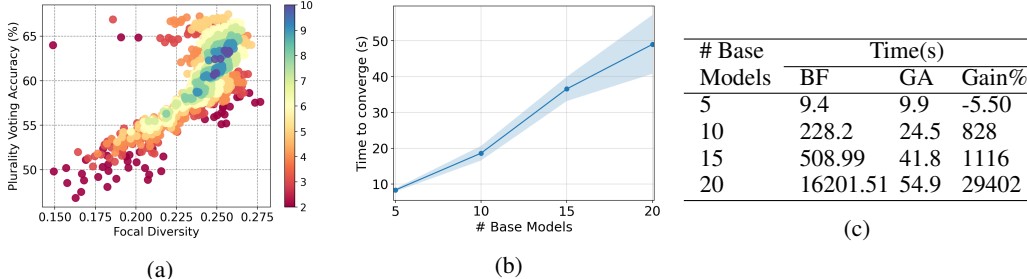

| # Base | Time(s) | | |
|--------|---------|------|--------|
| Models | BF | GA | Gain% |
| 5 | 9.4 | 9.9 | -5.50 |
| 10 | 228.2 | 24.5 | 828 |
| 15 | 508.99 | 41.8 | 1116 |
| 20 | 16201.51 | 54.9 | 29402 |

(a)             (b)             (c)

Figure 4: Focal diversity optimized ensemble pruning with brute force or Genetic Algorithm (GA)

ble sets with the colors representing the size of the ensemble sets for a pool of 10 base models on *mini*ImageNet. Each dot represents a subset of the pool of $M = 10$ base models. We observe from Figure 4a a clear correlation between focal diversity and ensemble accuracy: as the focal diversity increases, the accuracy of the ensemble increases. Moreover, ensemble teams of small size show high diversity and high accuracy compared to ensemble of large size. For example, in Figure 4a, the best-performing ensembles with high focal diversity typically have 2-4 base models. This indicates that with our focal diversity-optimized ensemble pruning, we can find high-performing ensembles of low complexity. Consider those dots on the top-right of Figure 4a, they are the sub-ensembles of small size with high focal diversity and high top-1 prediction accuracy. Thus, we create a pruning score metric by taking the convex combination of the diversity and accuracy of each ensemble set, i.e., $s_i = w_1 a_i + w_2 \lambda_i$. While $a_i, \lambda_i \in [0, 1]$ represent accuracy and focal diversity scores for ensemble set $\mathcal{E}_i$. The weights $w_1 + w_2 = 1$ represent the importance that one can put on the metrics to calculate the pruning scores. This allows us to create an ensemble selection strategy that focuses more on diversity and less on accuracy and vice versa.

**Speeding up ensemble pruning with genetic algorithm.** In the beginning of this section, we mention the exponential growth of the number of candidate ensemble sets, as $M$ gets larger, for ensembling from a pool of $M$ base models. For example, for $M = 20$, we must compute the focal diversity score for $1,048,555$ candidate ensemble sets. Figure 4b shows the time of convergence for the Genetic Algorithm (GA) (Mirjalili & Mirjalili, 2019) against the number of base models in the pool. Figure 4c shows the comparison between Brute Force (BF) implementation and GA. From Figure 4c, the Brute Force (BF) approach to compute the focal diversity and the plurality-based accuracy for each candidate ensemble of size $m$ ($1 \leq m \leq M$) is expensive. This motivates us to implement a Genetic Algorithm, which takes significantly less time to reach the best combination, as shown in Figure 4b and Figure 4c. The Genetic algorithm requires (i) the representation of a candidate solution and (ii) a fitness function to evaluate the solutions. As shown in Figure 5, we use the binary vector, where each index represents the presence of the base model in the ensemble set, to represent a solution $\alpha_i$. We use our pruning score calculation as the fitness function, i.e., $r(\alpha_i) = w_1 a_i + w_2 \lambda_i$. The initial population contains randomly created candidate solutions. During selection, the most fitted solutions survive to the next population. As the last step, we reproduce

new solutions by performing a cross-over among the best-fitted solutions. The procedure is repeated until we reach a plateau or a predetermined fitness function value. In Figure 4b, for 20 different base models, we reach the best set under a minute.

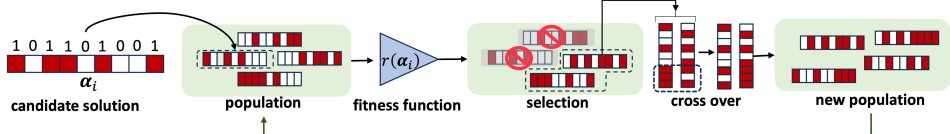

Figure 5: We show the Genetic Algorithm procedure.

## 5 EXPERIMENTS

**Ensemble Fusion Performance** Table 1 shows the ensemble fusion performance of FUSIONSHOT using five backbone algorithms (see columns) and five metric-space distance functions (see rows) on *mini*-Imagenet dataset. In row-wise, we show that using the fusion channel for feature extraction ensemble, FUSIONSHOT achieves the top-1 performance improvement by 7.69 − 9.78% over the classical few-shot models, and 4% for Simpleshot. In column-wise, we show that using the fusion channel for latent distance ensemble, we get up to 4% top-1 performance improvements.

| Method | Dist. | Conv4 | Conv6 | ResNet10 | ResNet18 | ResNet34 | EnsBB | Gain |
|---|---|---|---|---|---|---|---|---|
| Matching | Cosine | $51.73_{0.75}$ | $47.95_{0.79}$ | $50.85_{0.84}$ | $50.89_{0.78}$ | $50.90_{0.84}$ | $\mathbf{56.79}_{0.45}$ | 9.78% |
| Prototypical | L2 | $48.42_{0.79}$ | $49.17_{0.79}$ | $53.01_{0.78}$ | $51.72_{0.81}$ | $53.16_{0.84}$ | $\mathbf{57.39}_{0.45}$ | 7.96% |
| MAML | MLP | $45.51_{0.77}$ | $47.13_{0.84}$ | $50.83_{0.84}$ | $47.57_{0.84}$ | $48.92_{0.83}$ | $\mathbf{54.74}_{0.44}$ | 7.69% |
| Relation | CNN | $48.57_{0.82}$ | $48.86_{0.81}$ | $49.26_{0.85}$ | $47.07_{0.77}$ | $48.30_{0.77}$ | $\mathbf{53.75}_{0.47}$ | 9.11% |
| Simpleshot | L2 | $48.90_{0.73}$ | $50.26_{0.75}$ | $61.38_{0.81}$ | $62.61_{0.80}$ | $61.96_{0.77}$ | $\mathbf{65.09}_{0.45}$ | 3.96% |
| EnsDistance | | $\mathbf{53.28}_{0.48}$ | $\mathbf{52.70}_{0.45}$ | $\mathbf{62.36}_{0.44}$ | $\mathbf{64.26}_{0.44}$ | $\mathbf{64.46}_{0.41}$ | | |
| Gain | | 3.00% | 4.85% | 1.60% | 2.64% | 4.03% | | |

Table 1: Few-shot ensemble fusion gains for both the fusion channel of five feature extraction backbone algorithms (columns) and the fusion channel of five latent distance methods (rows)

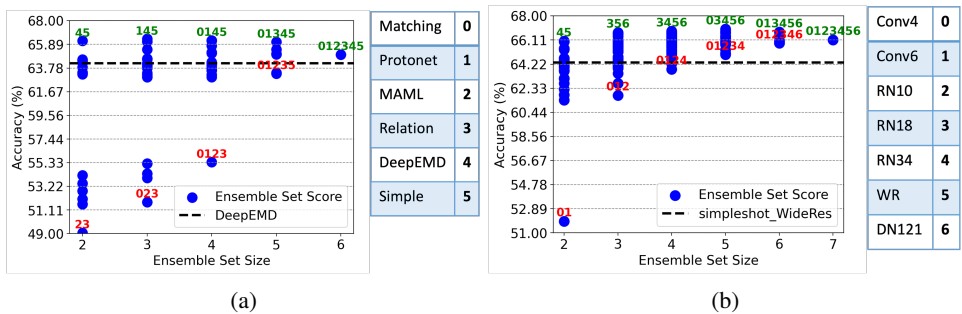

(a)                                                                          (b)

Figure 6: Performance of 1-shot 5-way ensembles (*mini*-Imagenet) produced (a) by latent distance fusion and (b) by feature extraction fusion. The green and red texts represent the best and worst performing sets among the candidate ensemble sets of the same team size. The horizontal line is the performance of the best base model.

**Ensemble Performance by Fusion of Latent Distance Methods.** In this set of experiments, we zoom in the column on ResNet18 in Table 1 to analyze the performance impact of ensemble fusion by combining multiple latent distance comparison methods. We add DeepEMD to the collection to get a total of six metric-space distance methods. We first train six few-shot models independently, each is jointly optimized with ResNet18 for feature extraction by embedding, and one of the six metric-space distance methods for query matching prediction. Figure 6a shows the performance measurement of the trained ensemble models over novel set of the *mini*-Imagenet dataset in the 1-shot 5-way few-shot setting. Figure 6a shows at the top the selection of the best performing candidate ensembles of varying team sizes. We observe that multiple ensemble teams outperform DeepEMD (Zhang et al., 2020), the best individual model in the pool of six latent distance models. Note that DeepEMD is the best individual model with top-1 accuracy of 63.9%. From Figure 6a, we observe that several ensembles in different team sizes can outperform the best member model − DeepEMD. First, the ensemble of Simpleshot and DeepEMD further improves the top-1 prediction over DeepEMD. By adding Protonet, the ensemble of the three models (145) outperforms most of

other ensembles, including the 2-model ensemble of Simplenet and DeepEMD, and the 6-model ensembes (012345), showing the effectiveness of our FUSIONSHOT focal diversity optimized ensemble pruning and our learn-to-combine parametric approach to ensemble fusion.

| Method | Backbone | Dist. Func. | Embedding Dimension | *mini*-Imagenet | |
|---|---|---|---|---|---|
| | | | | 1-shot | 5-shot |
| Simpleshot | DenseNet121 | L2 | 1024 | $63.53_{0.79}$ | $79.11_{0.55}$ |
| Simpleshot | WideResNet | L2 | 640 | $64.34_{0.79}$ | $78.80_{0.59}$ |
| DeepEMD | ResNet12 | EMD | $640 \times 5 \times 5$ | $64.21_{0.75}$ | $80.51_{0.54}$ |
| TADAM | ResNet12 | L2 | 640 | $58.50^{\dagger}_{0.30}$ | $76.70^{\dagger}_{0.30}$ |
| FEAT | WideResNet | Cosine | 640 | $61.72^{\dagger}_{0.11}$ | $78.49^{\dagger}_{0.15}$ |
| LEO | WideResNet | KLDiv | 640 | $61.76^{\dagger}_{0.08}$ | $77.59^{\dagger}_{0.12}$ |
| Robust-20 | ResNet18 | Cosine | 512 | $63.73^{\dagger}_{0.62}$ | $81.19^{\dagger}_{0.43}$ |
| **FusionShot**$^{\text{dist}}$: ResNet18 $\times$ (L2, L2, EMD) | | | 512 | $\mathbf{66.38}_{0.10}$ | $\mathbf{81.58}_{0.36}$ |
| **FusionShot**$^{\text{bb}}$: (RN18-34, DN121, WR, CN4)$\times$ L2 (512, 1024, 640, 1600) | | | | $\mathbf{66.97}_{0.10}$ | $\mathbf{81.12}_{0.36}$ |

(a)

| Method | Backbone | Dist. Func. | CUB | | *mini*-Image→CUB | |
|---|---|---|---|---|---|---|
| | | | 1-shot | 5-shot | 5-shot (blind) | 5-shot |
| Matching | ResNet18 | Cosine | $73.49_{0.89}$ | $83.64_{0.60}$ | $52.17_{0.74}$ | $53.07^{\dagger}_{0.74}$ |
| Prototypical | ResNet18 | L2 | $72.07_{0.93}$ | $85.01_{0.60}$ | $55.24_{0.72}$ | $62.02^{\dagger}_{0.74}$ |
| Relation | ResNet18 | CNN | $68.58_{0.94}$ | $82.75_{0.58}$ | $50.93_{0.73}$ | $57.71^{\dagger}_{0.73}$ |
| MAML | ResNet18 | MLP | $68.42_{1.07}$ | $82.70_{0.65}$ | $46.85_{0.72}$ | $51.34^{\dagger}_{0.72}$ |
| Simpleshot | ResNet18 | L2 | $66.63_{0.88}$ | $82.63_{0.65}$ | $67.38_{0.70}$ | - |
| DeepEMD | ResNet12 | EMD | $74.39_{0.85}$ | $87.65_{0.55}$ | $77.44_{0.70}$ | $78.86^{\dagger}_{0.65}$ |
| Robust-20 | ResNet18 | L2 | - | - | - | $65.04^{\dagger}_{0.57}$ |
| **FusionShot**$^{\text{dist}}$: ResNet18 $\times$ (L2, L2, EMD) | | | $\mathbf{78.64}_{0.38}$ | $\mathbf{88.95}_{0.29}$ | $\mathbf{78.02}_{0.38}$ | |

(b)

Table 2: Comparison with existing SOTA methods on 1-shot 5-way and 5-shot 5-way performance using (a) *mini*-Imagenet and CUB and (b) cross-domain scenario. FusionShot$^{\text{dist}}$ is the distance fusion of Protonet, Simpleshot, and DeepEMD with ResNet18 backbone architecture. FusionShot$^{\text{bb}}$ is the backbone fusion of Conv4, ResNet18-34, WideResNet, and DenseNet121 with Simpleshot as the metric-space distance method. Data with the $^{\dagger}$ symbol is taken from the corresponding work.

**Ensemble Fusion by Backbone Feature Extractors.** The next set of experiments is conducted to measure the effectiveness of ensemble fusion by combining multiple backbone algorithms for feature extraction under one fixed latent distance comparison method. We choose Simpleshot in this set of experiments and paired with one of the seven backbone DNN architectures: Conv4, Conv6, ResNet10, ResNet18, ResNet34, WideResNet and DenseNet121. We first independently train seven few-shot models by using Simpleshot as the metric-space distance method, jointly optimized with one of the seven backbone DNN architectures for embedding based feature extraction. From Figure 6b, we make two observations. (1) A large number of the ensemble teams show higher performance than the best-performing base model, which is SimpleShot with WideResNet. (2) The best-performing ensemble teams, highlighted in green, have highfocal diversity and the most diverse backbone architectures, e.g., the best performing 3-model ensemble {356} is composed of ResNet18, DenseNet121 and WideResNet.

**Comparison with existing SOTA Few-shot methods** We choose the six latent distance comparison methods with ResNet18 as the fixed backbone architecture for feature extraction, as illustrated in Figure 6b. By focal diversity ensemble pruning, we obtain Protonet, SimpleShot, and DeepEMD with ResNet18 as the best ensemble by distance fusion, denoted by FusionShot$^{\text{dist}}$. Similarly, we choose the seven DNN backbone architectures as shown in Figure 6a and obtain SimpleShot with Conv4, ResNet18-34, DenseNet121, and WideResNet as our best ensemble of four models, denoted by FusionShot$^{\text{bb}}$. We include several recent representative SOTA algorithms in Table 2, including TADAM (Oreshkin et al., 2018), FEAT (Ye et al., 2020), LEO (Rusu et al., 2018), and Robust-20 (Dvornik et al., 2019). The ensemble models recommended by our focal-diversity optimized ensemble pruning improves the top-1 prediction performance by up to 4%, compared to the other methods on both *mini*-Imagenet and CUB datasets for 5-way 1-shot and 5-way 5-shot scenarios.

**Interpretation of Ensemble Fusion.** This set of experiments is designed to provide intuitions to interpret the performance improvement of FUSIONSHOT ensemble fusion. Figure 7a compares the

performance of each base model with the FUSIONSHOT for each of the 20 novel classes of mini-Imagenet. We make three observations. (i) The performance of FUSIONSHOT ensemble fusion can serve as the upper bound for each base model. (ii) Protonet and Simpleshot show similar high-low performance patterns in the set of novel classes. Although unlike Protonet, Simpleshot uses the k-nearest neighbor distance comparison method, both use L2 distance as their vector similarity metric. (iii) DeepEMD displays nearly uniform performance across all novel classes, likely due to the use of earth movers' distance (EMD). Figure 7b reports the results of another set of experiments. Here, we analyze all the base model predictions in 45000 episodes and count the errors that each model made individually or together. The red bars show the number of errors that each individual makes or the number of errors made by all models in the 2-model or 3-model combinations, while the green bar shows the number of episodes that are corrected by our focal diversity selected ensemble for top-1 prediction performance. We make three observations. (1) The right-most red bar shows that the ensemble of 3-distance methods (Protonet, Simpleshot, and DeepEMD) with ResNet18 backbone architecture failed together in 8270 episodes, and our ensemble fusion model can successfully correct 97 of them (1.2%). (2) Consider the 2-model ensemble {1-2}, there are 11112 episodes that both Simpleshot and DeepEMD made incorrect top-1 prediction, and our ensemble fusion method can correct 891 of them (8%). Finally, for Protonet (left-most), our FUSIONSHOT ensemble can correct 9178 out of 21728 incorrect episodes, offering over 42% performance improvement.

**Cross-domain Performance** This set of experiments evaluates the cross-domain performance of our ensemble of base models by first training on the *mini*-Imagenet and then testing on the CUB dataset. The ensemble consists of the base models trained on mini-ImageNet and never sees the new CUB dataset during meta-learning. The CUB dataset is only in the novel set for testing. This *blind* setting simulates closely the real-world scenarios. Table 2b shows the performance of FusionShot in blind setting. Even though there is a 10% gap between the first two best-performing base models, FUSIONSHOT shows comparable performance with the best base model. When we remove Deep-EMD, FUSIONSHOT improves the best base model performance by up to 6% (see Appendix). We also show that FUSIONSHOT outperforms Robust-20 (Dvornik et al., 2019), an ensemble method using 20 ResNet18 models, which is not trained blindly for cross-domain scenario.

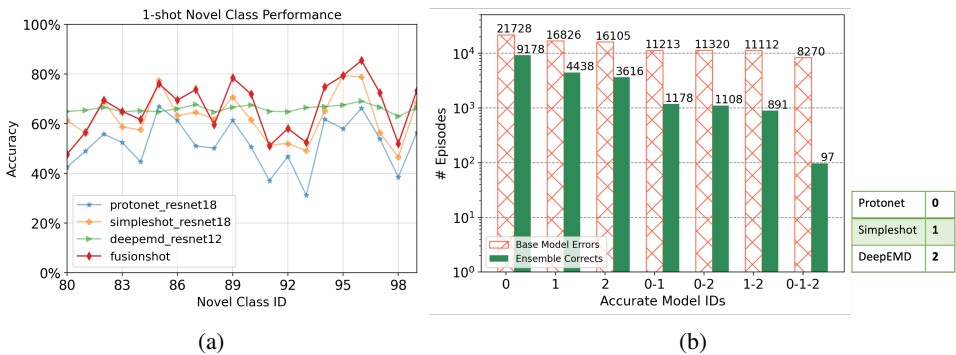

|  | (a) |  | (b) |

Figure 7: (a) The accuracy of base models vs FUSIONSHOT on each novel class out of 45000 episodes. (b) Red bar: # errors made by single base model or all models in a team out of 45000 novel episodes (1-shot 5-way, *mini*-Imagenet). Green bars: # corrected episodes by FUSIONSHOT.

## 6 CONCLUSION

We have presented a focal diversity optimized few-shot ensemble learning approach, coined as FU-SIONSHOT. First, we explore and compare two alternative fusion channels to ensemble multiple few-shot models. One is the fusion of various latent distance comparison methods for distance based loss optimization. The other is the fusion of multiple backbone DNN algorithms to learn and extract latent features differently. Moreover, we introduce a focal-diversity optimized few-shot ensemble pruning method for further boosting the performance of ensemble fusion. Extensive experiments are conducted on popular few-shot benchmarks (mini-Imagenet and CUB) with three learning scenarios: object recognition, fine-grained image classification, and cross-domain classification. The results show that our ensemble fusion approach can select the best performing ensembles from a pool of base few-shot models, which outperform both the representative SOTA models and the best base model used for composing the ensemble.

# 7   REPRODUCIBILITY STATEMENT

We make the following effort to enhance the reproducibility of our results.

- For FUSIONSHOT implementation, a link to an anonymous downloadable source is included in our abstract. We also briefly introduce the implementation of FUSIONSHOT in Appendix C2.

- We show a brief description of the representative SOTA few-shot models in Appendix C1. Detailed settings and the hyper-parameter selection logistics can be found in Appendix C2.

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

**Organization of Appendix**

APPENDIX

This appendix includes additional materials that elaborate the discussions in the main paper. We organize the materials by using the same section name from Section 2 (Appendix Section B) to Section 4 (Appendix D) to facilitate the reading. We also include a brief overview of the few-shot recent literature in Appendix E.

## A FEW-SHORT LEARNING: THE REFERENCE ARCHITECTURE

### A.1 TRAINING, VALIDATION AND NOVELTY TESTING

In contrast to classical supervised learning, a few-shot learning model is trained, validated and tested on strictly disjoint partitions of labeled data in terms of classes. Put differently, the classes in training set, validation set and novel set are also disjoint. For example, the *mini*-Imagenet dataset consists of 100 classes with 600 samples each. To train a few-shot learner, we split the 100 classes into 64, 16, and 20 disjoint partitions for training, validation, and novel test sets, containing 38400, 9600, and 12000 samples respectively. We can formulate the problem as follows: Let $\mathcal{D} = \{(\mathbf{I}_i, y_i)\}_{i=1}^{L}$ denote a dataset with each sample $\mathbf{I}_i$ paired with a corresponding class label $y_i$ and $y_i \in C = \{c_1, c_2, \ldots, c_L\}$. Note that instead of pairing with a label, one can also pair with another input for self-supervised learning, such as a pair of food image and recipe text (Xie et al., 2021). To train a few shot learner, we first partition $\mathcal{D}$ into $\mathcal{D}^{\text{train}}$, $\mathcal{D}^{\text{val}}$, and $\mathcal{D}^{\text{novel}}$, denoting training set, validation set, and novel test set respectively, and $L^{train} + L^{val} + L^{novel} = L$. Let $B$ denote the number of samples per class. We have $|\mathcal{D}^{\text{train}}| = L^{train} \times B$, $|\mathcal{D}^{\text{val}}| = L^{val} \times B$, and $|\mathcal{D}^{\text{novel}}| = L^{novel} \times B$.

### A.2 EPISODE-BASED META-LEARNING AND META-TESTING

The few-shot learning consists of two stages: meta-learning and meta-testing. In meta-learning, we train the model by using $\mathcal{D}^{\text{train}}$, and validate its performance on the $\mathcal{D}^{\text{val}}$, including tuning hyper-parameters. In meta-testing, the performance of the trained few-shot model will be tested on $\mathcal{D}^{\text{novel}}$. Figure 2 gives a sketch of the *episode* based few-shot learning architecture (Vinyals et al., 2016). We call one forward pass as an episode. For *K-way N-shot* learning ($1 \leq K << min\{L^{train}, L^{val}, L^{novel}\}, 0 \leq N << B$), an episode consists of a query $\mathbf{Q}$ and a support set $\mathcal{S}$ of $K$ classes with $N$ samples per class. Both $\mathbf{Q}$ and $\mathbf{S}$ are randomly sampled from $\mathcal{D}^{\text{train}}$ or $\mathcal{D}^{\text{val}}$ or $\mathcal{D}^{\text{novel}}$ to create the training episode set, the validation episode set, or the novel episode set for testing respectively. The composition of an episode should meet the following constraints: First, $\mathcal{S} = \{\{\mathbf{I}_1^{c_{i1}}, \ldots, \mathbf{I}_N^{c_{i1}}\}, \ldots, \{\mathbf{I}_1^{c_{iK}}, \ldots, \mathbf{I}_N^{c_{iK}}\}\}$, and $|\mathcal{S}| = N \times K$. Let $\mathbf{I}_i^{c_{ij}}$ denote an input sample $i$ belonging to class $c_j$. After creating the support set $\mathcal{S}$, a query $\mathbf{Q}$ is randomly sampled to pair with $\mathcal{S}$ for composing the episode. The query $\mathbf{Q}$ should meet the following criteria: (i) $\mathbf{Q}$ should be drawn from samples in the training set, i.e., $\mathbf{Q} \in \mathcal{D}^{\text{train}} \setminus \mathcal{S}$; and (ii) $\mathbf{Q}$ should refer to a sample that does not exist in the support set $\mathcal{S}$ but belongs to one of the $K$ classes used in $\mathcal{S}$, i.e., $\mathbf{Q} \notin \mathcal{S}, \mathbf{Q} = \mathbf{I}^{c_{ij}}$ $c_{ij} \in \{c_{i1}, \ldots, c_{iK}\}, \mathcal{S} \subset \mathcal{D}^{\text{train}}$. The number of episodes used per epoch for few-shot learning on both mini-Imagenet and CUB benchmark is 1600, 1600, and 600 for training, validation and novelty testing respectively. Episode based training for a $K$-way $N$-shot learner will iterate through all training episodes and the total number of iterations is a hyper-parameter, usually larger than one epoch, and is set differently by different few-shot algorithms (see Appendix for detail).

For each iteration, we send one episode $(\mathcal{S}, \mathbf{Q})$ to the DNN embedding function $f_\theta(.)$ to generate a total of $|\mathcal{S}| + 1$ latent embedding mappings, denoted by $\mathcal{H}_i = \{\mathbf{h}_1^{c_i}, \ldots, \mathbf{h}_N^{c_i}\}$, where $\mathbf{h} = f_\theta(\mathbf{I}^{c_i})$ $(i = 1, \ldots, K)$, plus one query embedding $\mathbf{h}_q = f_\theta(\mathbf{Q})$, as shown in the middle of Figure 2. Second, the latent distance function $g_\psi(.)$ takes all $N$ latent embeddings of the same class $c_i$ to obtain a per-class integrated latent embedding by taking the averaging, for each of the $K$ classes in the support set, denoted by $\mathcal{S} \{\mathbf{e}_{c_1}, \ldots \mathbf{e}_{c_i}, \ldots, \mathbf{e}_{c_K}\}$. Then we use a latent-distance-method $z = d_\psi(\mathbf{h}_q, \mathbf{e}_{c_i})$ to compute the distance between query embedding $\mathbf{h}_q$ and each of the $K$ class embedding $\mathbf{e}_{c_i}$. Followed by *softmax* and the entropy based loss, we obtain the meta-loss and perform stochastic gradient decent and back-propagation to initiate the next iteration of the learning with a new episode. This training process repeats until all training episodes are consumed. The latent-distance-method $d_\psi(.)$ is typically a neural network based on $L2$ or $Cosine$ (Sung et al., 2018; Vinyals et al., 2016; Koch et al., 2015; Snell et al., 2017). Thus, the DNN embedding function $f_\theta(.)$ with parameters $\theta$ and the latent similarity function $g_\psi$ with parameters $\psi$ are jointly trained.

### A.3 Pre-training before Few-shot Meta-training

*Pretraining* is one of the options to further improve the transfer-ability and adaptability of few-shot learning to **unseen** novel data. Instead of beginning with randomly initialized weights, this optimization will choose to pre-train the DNN embedding based feature extractor $f_\theta(.)$ on the training set by adding a supervised classifier layer. Let $\tilde{y}_i = \text{softmax}(\mathbf{W}^T f_\theta(\mathbf{I_i}))$ represent the predictions based on the jointly trained model, where $\mathbf{I}_i \in \mathcal{D}^{\text{train}}$ and $\mathbf{W}^T \in \mathbb{R}^m \times z$ is the classifier layer, $m$ denotes the embedding dimensions of the feature extractor and $L_{train}$ denotes the number of classes in $\mathcal{D}^{train}$. This approach jointly trains the embedding with the cross-entropy loss to minimize the classification error:

$$\mathcal{L}_{\log}(\theta, \mathbf{W}) = -\sum_{k=1}^{z} \tilde{y}_k log(p(\tilde{y} = k \mid \mathbf{I}_i; \theta, \mathbf{W}))$$

## B Design Overview of FusionShot

### B.1 Ensemble Fusion through Learning

In FUSIONSHOT, we first determine the fusion channel to use for creating a pool of $M$ base models ($M$ is a configurable hyper-parameter). Then we will construct the FUSIONSHOT specific training dataset to train the FUSIONSHOT model for effective learning of few-shot ensemble fusion. We learn a few-shot ensemble model in four steps. (1) We choose $M$ independently trained few-shot models as the pool of $M$ base models. For example, the models can be trained using one or $M$ different backbone algorithms for feature extraction embedding and $M$ different latent distance methods, as shown in Figure 3. (2) We feed the training episodes, one at a time, to the $M$ component models. For each episode $i \in E_{train}$, denoted by $(\mathbf{Q}_i, \mathcal{S}_i)$, we collect the query prediction probability vector of size $K$, denoted by $(\hat{\mathbf{y}}_i^1, \ldots, \hat{\mathbf{y}}_i^K)$, each of the $K$ values corresponds to the confidence of matching the query $\mathbf{Q}_i$ to one of the $K$ classes in the support set $\mathcal{S}_i$. Let $L_{train}$ denote the number of training episodes, and $M$ is the total number of base few shot models. We have the ensemble fusion training set $\mathcal{Y}^{\text{train}} = \{\hat{\mathbf{y}}_i^j\}_{i=1, j=1}^{M, L_{train}}$. (3) We feed the ensemble fusion training data collected for learning to combine using a MLP network with cross-entropy loss optimization, as illustrated in Figure 3. (4) We use the validation episodes and novel episodes to create $\mathcal{Y}^{\text{val}}$ and $\mathcal{Y}^{\text{novel}}$. We use the predictions of the novel dataset, $\mathcal{Y}^{\text{novel}}$, as the performance evaluation of each ensemble model in FUSIONSHOT.

**Query prediction probability vector.** For each base model of $K$-way $N$-shot, the forward pass in meta-learning (recall Figure 2) will be used to collect the episode-based prediction probability vector, $(\hat{\mathbf{y}}_i^1, \ldots, \hat{\mathbf{y}}_i^K)$, using the training episode set, For episode $i \in E^{\text{train}}$, the probability vector is of size $K$, and each of the $K$ probability values indicates the confidence of matching $\mathbf{Q}_i$ to one of the $K$ categories in the support set $\mathcal{S}_i$ based on their latent features. The top-1 prediction for $\mathbf{Q}_i$ is the category with the highest probability value. The average top-1 performance is the average of the top-1 probability over the set of novel episodes ($E^{\text{novel}}$). Therefore, given the set of training episodes, $E^{\text{train}}$, the probability vector is obtained by maximizing the likelihood of $\phi$:

$$l(\phi) = \arg\max_{\phi} \prod_{i=1}^{E^{\text{train}}} p_\phi(y|\mathcal{S}_i, \mathbf{Q}_i), \tag{2}$$

where $\phi$ denotes the model parameters that parameterize the probability function over each training episode $i \in E^{\text{train}}$. Note that the backbone embedding function (parameters $\theta$) is jointly trained with the latent distance function (parameters $\psi$) by maximizing the equation 2, which can reduce to minimize the *cross-entropy* loss on the model with parameters $\phi$ and the probability $p_\phi(y|\mathcal{S}_i, \mathbf{Q}_i)$ over the support set of $K$ classes in each episode $i \in E^{\text{train}}$:

$$\mathcal{L}_{\log}(\phi) = \arg\min_{\phi} -\sum_{i=1}^{E^{\text{train}}} y \log p_\phi(y|\mathcal{S}_i, \mathbf{Q}_i) \tag{3}$$

Each iteration the parameters are updated using Stochastic Gradient Descent (SGD). Instead of the cross-entropy loss (equation 3), one may also use other loss functions, e.g., MSE (Sung et al., 2018).

Let $\mathbf{W}_\gamma, b_\gamma$ denote the learnable model parameters for the MLP (learn to combine) model. For each ensemble of size $m$ ($1 \leq m \leq M$), after each of the $m$ base models (all $K$-way $N$-shot) produce a query prediction vector, we concatenate the predictions of the ensemble of $m$ base models, and the model parameters are $\mathbf{W}_\gamma \in \mathbb{R}^{m \times K}$. At the final step, the model performs softmax to produce

the probability for each of the $K$ classes in the support set, as illustrated in Figure 3. The ensemble learner will output the ensemble fusion optimized prediction based on the logits layer of the MLP (learn to combine) model:

$$\tilde{\mathbf{y}}^{fusion} = \text{softmax}(\mathbf{W}_\gamma^T[\hat{\mathbf{y}}^1, \ldots, \hat{\mathbf{y}}^P] + b_\gamma), \tag{4}$$

### B.2 SIZE SENSITIVE FOCAL DIVERSITY PRUNING

In this section, we present a further formulation of our fitness function shown in section 4.2. There is no limit on the size of the ensemble set in theory, however, there are practical bounds on the number of ensemble models one can use. Thus, we present a fitness function formulation that is set-size-sensitive:

$$s_i = w_1 a_i + w_2 \lambda_i - \beta \frac{m}{M} \tag{5}$$

Here we introduce the $m/M$ term to penalize the fitness score based on the size of the ensemble set, $m$, to the total number of available models $M$. To control the amount of importance one can put on the ensemble size we multiply the penalty term with $\beta$ value.

We test the effectiveness of this function on the largest model pool with $M = 28$ and the model combinations are as follows: (Matching, Proto, Relation, MAML, Simpleshot) $\times$ (Conv4, Conv6, Resnet10, Resnet18, Resnet34), Simpleshot $\times$ (WideResNet, DenseNet121), DeepEMD. Note that the number of combinations one can produce is above 200 million. With our Genetic Algorithm-powered Focal Diversity Pruning algorithm we pruned the sets in 66.31 seconds. We get a result with size of 5 when we set the $\beta = 0.1$ and a result with size of 4 when we set the $\beta = 0.2$. The resulting sets with the performance is as follows:

- (MAML-RN18, MAML-RN34, SimpleShot-WR, DeepEMD) $\rightarrow 66.39 \pm 0.42$
- (PN-RN18, MAML-RN18, MAML-RN34, SimpleShot-WR, DeepEMD) $\rightarrow 66.29 \pm 0.44$

We used (Gad, 2021) library to implement Genetic Algorithm.

## C EXPERIMENTS

### C.1 EXPERIMENTAL SETUP

**Datasets and Scenarios:** To evaluate FUSIONSHOT against the existing SOTA few-shot models, we use three different learning scenarios: object recognition, fine-grained image classification, and cross-domain classification, as suggested by (Chen et al., 2019). We use *mini*-ImageNet benchmark for the first scenario. The dataset contains 100 classes from ImageNet, 600 samples per class. The partition of classes for training, validation, and novel datasets is random. For ease of comparison, we follow the standard distribution suggested in (Ravi & Larochelle, 2016). For the second scenario, we use the Caltech-UCSD Birds-200-2011 (CUB) dataset, containing 11,788 samples belonging to 200 classes and split it into 100 training, 50 validation, and 50 novel classes following (Hilliard et al., 2018). The literature calls the second scenario a fine-grained image classification because the goal of CUB is to distinguish the species of birds rather than what the object is, such as human or animal. For the cross-domain scenario, the standard approach is to train the model on one dataset and evaluate it on another, e.g., *mini*-ImageNet→CUB. The goal is to discover the effects of domain shifts to the few-shot models. In the *standard* approach, however, the model experiences the new dataset's validation classes during training. Specifically, the model stops training when the performance on the new dataset's validation classes decreases. In this paper, we use the *blind* cross-domain scenario, where the model sees the new dataset's classes only in novel dataset testing. The goal is to completely remove dependency on the training process and establish a consensus among few-shot learners using the ensemble learning method.

**Evaluation.** We perform our experiments with the standard settings in few-shot learning, which are 5-ways 1-shot and 5-ways 5-shot classification ($N = \{1, 5\}, K = 5$). Since few-shot learners perform episodic learning where each episode chooses its samples randomly, we report the mean accuracy (%) and 95% confidence interval of the novel set, $\mathcal{Y}^{\text{novel}}$, with 600 episodes for all scenarios.

**Implementation Details.** We first perform meta-training on all $M$ base models of an ensemble set using training and validation datasets. We then perform few-shot inference on the training, validation, and novel sets to create the inferece set for ensemble fusion, including to create the MLP for

learning to combine: $\mathcal{Y}^{\mathrm{train}}$, $\mathcal{Y}^{\mathrm{val}}$, and $\mathcal{Y}^{\mathrm{novel}}$. Since the episode creation process contains randomization, we must set the seed and keep the indexes of images during the inference so that inference sets contain predictions of the base models on the same set of support and query images. Finally, we train a neural network having two fully-connected hidden layers with 100 neurons, on the inference prediction set for 300 epochs, with the sigmoid activations between layers, and the Adam optimizer with a learning rate of 0.001. We will use seven backbone DNN architectues for evaluating the fusion of feature extraction. They are Conv4-6 (Vinyals et al., 2016), ResNet10, ResNet12, ResNet18, ResNet34, WideResnet and DenseNet121. For the fusion of multiple meta-methods for latent distance comparison, we consider six alternative parametric methods, such as ProtoNet, MatchingNet, RelationNet, Model-Agnostic Meta-Learning (MAML) model, DeepEMD and Simpleshot. For all the base models, we use the standard augmentation during training, incl., random crop and color augmentation. While the classical methods do not apply any feature adaption method, DeepEMD and Simpleshot methods perform pre-training on their backbones using the training set. Depending on the backbone architecture, the size of the image fed to the model changes between $224 \times 224$ and $84 \times 84$. The details on the size, speed, and complexity of each base model are provided in the next subsection.

## C.2 BASE MODEL DETAILS

| Method | #Train Episodes | Input Dim. | Train Aug. |
|---|---|---|---|
| protonet | 60,000 | $3 \times 224 \times 224$ | Resize, CenterCrop, Normalize |
| MatchinNet | 60,000 | $3 \times 224 \times 224$ | Resize, CenterCrop, Normalize |
| RelationNet | 60,000 | $3 \times 224 \times 224$ | Resize, CenterCrop, Normalize |
| MAML | 60,000 | $3 \times 224 \times 224$ | Resize, CenterCrop, Normalize |
| DeepEMD | 5,000 | $3 \times 84 \times 84$ | RandomResizedCrop, RandomHorizontalFlip, Normalize |
| SimpleShot | 9,000 | $3 \times 224 \times 224$ | Resize, CenterCrop, Normalize |

(a)

| Method | Distance Function | Pre-Train Loss | Meta-Loss Function | Optimizer |
|---|---|---|---|---|
| protonet | Euclidean | - | Cross Entropy | Adam |
| matchingnet | Cosine | - | Cross Entropy | Adam |
| relationnet | CNN | - | MSE | Adam |
| maml | MLP | - | Cross Entropy | Adam |
| deepemd | EMD | Cross Entropy | Cross Entropy | SGD |
| simpleshot | KNN (Cosine) | Cross Entropy | - | SGD |

(b)

| Method | Backbone | Embed. Dim. | Pretrain | Episode Time | Total Size |
|---|---|---|---|---|---|
| Protonet | ResNet18 | 512 | No | 29ms | 42.672 MB |
| MatchinNet | ResNet18 | 512 | No | 60ms | 70.719 MB |
| RelationNet | ResNet18 | 512 | No | 28ms | 69.707 MB |
| MAML | ResNet18 | 512 | No | 161ms | 42.682 MB |
| DeepEMD | ResNet18 | $640 \times 5 \times 5$ | Yes | 354ms | 47.431 MB |
| SimpleShot | ResNet18 | 512 | Yes | 94ms | 42.768 MB |

(c)

Table 3: We show the training setting of each base model in (a) and (b). The cost of each model in terms of spatial and temporal is shown in (c)

In this section, we give details on our models, where each of them utilizes a different latent distance function. For all the extracted query and support embeddings we use the notation of $f_\theta(\mathbf{Q}) = \mathbf{h}_q$ and $f_\theta(\mathbf{I}_i) = \mathbf{h}_i$, where $\mathbf{I}_i \in \mathcal{S}$ and $\theta$ is the backbone parameters. On below we show that $\hat{y}_k$ is the $k^{th}$ value of probability vector $\hat{\mathbf{y}}_k$.

### C.2.1 PROTOTYPICAL NETWORKS

After the backbone architecture produces each embedding, Prototypical Networks take the average of embeddings that are in the same class and call them class prototypes. Then, they classify the query

by looking at the nearest Euclidian distance from the query embedding to the class prototypes. When all the distances are calculated, a probability value is calculated for each class as follows:

$$p(\hat{y}_k|\mathbf{Q}, \mathcal{S}) = \frac{exp(-||\mathbf{h}_q - \mathbf{e}_k||_2^2)}{\sum_{k'} exp(-||\mathbf{h}_q - \mathbf{e}_{k'}||_2^2)}, \tag{6}$$

where we use the same notation $\mathbf{e}_k$ for class prototypes which we show earlier in Appendix section A.1. Prototypical Networks suffer the cross-entropy loss.

### C.2.2   MATCHING NETWORKS

Matching networks, however, in the simplest form calculate softmax over the cosine distances between the query and all the support embeddings, $\alpha(\mathbf{h}_q, \mathbf{h}_i) = \frac{exp(cos(\mathbf{h}_q, \mathbf{h}_i)))}{\sum_j^{|\mathcal{S}|} exp(cos(\mathbf{h}_q, \mathbf{h}_j))}$ which they call an attention value to the class. Then, Matching Networks perform a linear combination of the support labels:

$$p(\hat{y}_k|\mathbf{Q}, \mathcal{S}) = \sum_{i=1}^{|\mathcal{S}|} \alpha(\mathbf{h}_q, \mathbf{h}_i)\mathbf{1}(y_i = k), \tag{7}$$

where $\mathbf{1}$ is the identity function which takes 1 corresponding to the calculated class probability. Matching Networks, also, suffer the cross-entropy loss.

### C.2.3   RELATION NETWORKS

Relation networks utilize another CNN architecture called 'relation module' to perform a comparison between the query and support embeddings. The relation module takes the concatenated query embedding and a support embedding to produce a relation score $r$ representing the relation between the query and the sample. In the case of multiple shots, the relation networks sum all the relation scores for individual classes. Differently, the relation networks suffer mean squared error loss where the matched pairs have similarity 1 and mismatched pairs have similarity 0.

### C.2.4   MAML

MAML is also a parametric model that employs a linear layer with parameters $\mathbf{W}$ and $\mathbf{b}$ on top of the backbone $f_\theta(.)$ following a softmax operation. Differently, in each episode, it performs a small number of learning steps on the given support set, starting from the initial parameters $(\mathbf{W}, \mathbf{b}, \theta)$. The inner loop performs supervised learning by grouping the images in the same class and labeling them. After the inner loop iteration, the model predicts the query as follows:

$$p(\hat{y}_k|\mathbf{Q}, \mathcal{S}) = \text{softmax}(\mathbf{b}' + \mathbf{W}'f_{\theta'}(\mathbf{Q})), \tag{8}$$

where $(\mathbf{W}', \mathbf{b}', \theta')$ are the updated parameters by the inner loop by suffering the cross-entropy loss. Normally, the model is trained by updating the second-order gradients from inner loop parameters into the initial parameters but first-order approximation is taken to reduce the amount of memory cost. Dissimilar to other methods, MAML aims to learn the best initialization parameters.

### C.3   SIMPLE SHOT

Simple Shot does not perform meta-training. Simpleshot, first, trains a classifier on top of the backbone using supervised labels by minimizing the cross-entropy loss, $l(\mathbf{W}f_\theta(\mathbf{I}), y)$. Secondly, it removes the classifier and performs meta-testing by assigning the closest support embedding to the query embedding, i.e., it performs nearest-neighbor classification:

$$\hat{y}_k = \underset{c_i \in \{c_1, ..., c_K\}}{\arg\min} d(\mathbf{h}_q, \mathbf{h}^{c_i}) \tag{9}$$

where $d$ either can be Euclidian or cosine distance. Simpleshot provides simple transformations on the embeddings by centering and L2 normalization.

### C.4 DeepEMD

Deep EMD, first, pre-trains its backbone architecture by following the process we show in Appendix A.3. Second, it removes the classifier layer and performs meta-training. Deep EMD employs Earth Mover's Distance function as their distance metric to the extracted features of query and support samples:

$$\text{EMD}(\mathbf{Q}, \mathbf{I}) = \sum_{i,j} \tilde{x}_{i,j} \zeta_{i,j}, \tag{10}$$

where $\mathbf{I} \in \mathcal{S}$ and $\tilde{x}$ is the maximum flow of sending query weights to support weights and $\zeta$ is the cost between weights. DeepEMD calculates the distance by solving a linear program. Thus, it calls an LP solver in each iteration. To perform end-to-end training on the backbone, it calculates the Jacobian. After propagating the gradients coming from the distance function, it suffers cross-entropy loss.

#### C.4.1 Base Model Implementation Details

In the implementation of the Protonet, Matchingnet, Relationnet, and MAML; we used the code provided by (Chen et al., 2019). For the DeepEMD and Simpleshot we used the code provided by (Zhang et al., 2020) and (Wang et al., 2019), respectively. We separately train each base model with the hyperparameters shown in Table 3a and 3b. For all the hyperparameters we suggest checking our library for the reader.

In terms of spatial and temporal cost, we provide the capacity of each model with a ResNet18 backbone, in Table 3c. Note that, ResNet18 has a size of 42.67MB alone. Secondly, we provided the duration of one forward pass for each model. In an online set-up where each base model runs in parallel, the bottle-neck model will be DeepEMD.

We show the meta-training algorithm in Algorithm 1. Note that in each epoch, we perform multiple iterations of episode creation and forward pass of the created episode. The classes, $K$, are selected randomly from the $\mathcal{C}_{train}$. Then, we sample multiple images to create queries and a support set from the corresponding samples of the classes. Note that, the sampled data is in $K \times (N + M)$ shape. We take the first $N$ columns as support set, and the other columns as queries. Thus, in each iteration, we have $M$ amount of query for each class. One forward pass on the model produces the logits, and we compare the logits with the class IDs. The class IDs, $y$, are integers showing the positions of the classes in the support set for each query. Note that, $y$ is the same for each iteration regardless of the class sampled from $\mathcal{C}_{train}$. Lastly, we compare the logits with the $y$ and obtain episode accuracy and loss.

Secondly, we show the inference algorithm in Algorithm 2 to obtain predictions of the trained model on $\mathcal{D}^{\text{train}}$, $\mathcal{D}^{\text{val}}$, and $\mathcal{D}^{\text{novel}}$. The episode creation process is the same as the training algorithm with the difference of selected classes for the samples, i.e., $\mathcal{C}^{\text{train}}$, $\mathcal{C}^{\text{val}}$, and $\mathcal{C}^{\text{novel}}$. Since $y$ is the same for each iteration regardless of which class it is sampled from, the process is class invariant, which makes the models few-shot learners.

### C.5 Base Model Performances on Novel Classes

We analyze and compare the performance of these base models in Figure 8. Hence, all these based models are using the same backbone architecture ResNet18 as shown in Figure 8. We make two interesting observations. First, by observing the performance increase and decrease, we can say that most of the base models follow similar patterns over the novel classes. However, the variance of the models' performances in a particular class is high. Second, DeepEMD shows more stable and overall better top-1 and top-5 performance across different novel classes. However, for some novel classes, Simpleshot clearly outperforms DeepEMD, such as novel class IDs 85, 89, 94, 96 for both top-1 and top-5 performance.

Despite the fact that the accumulative behavior of base models shows similar patterns in terms of top-1 and top-5 performance ups and downs, Figure 9 shows that their individual episode decisions tend to provide different probability densities across the novel classes in the corresponding support sets. The ensemble fusion method can effectively refine the top-1 and top-5 predictions by not only taking into account of the highest scores of the models but also resolving the top-1 and top-5 prediction inconsistencies among the component base models of the ensemble.

---

**Algorithm 1:** Training Algorithm

---

**Input** : $N \leftarrow 1$ ;                         `// number of shots`
1   $K \leftarrow 5$ ;                          `// number of ways`
2   $M \leftarrow 15$ ;               `// number of query samples per class`
3   $\mathcal{D} \leftarrow \mathcal{D}_b$ ;                  `// the base dataset`
4   num_epoch ;                  `// number of epochs`
5   num_iter ;              `// number of iterations per epoch`
**Output:** Accuracy (acc) over time
6   **for** *epoch* **to** *num_epoch* **do**
7      Initialize acc $\leftarrow []$ ;           `// list to store accuracies`
8      **for** $i$ **to** *num_iter* **do**
9          Randomly select $\mathcal{C}_{train}$ classes: cls $\leftarrow$ random_select($\mathcal{C}_{train}, K$) ;  `// select random K`
              `classes from base classes`
10         Initialize data $\leftarrow []$ ;            `// list to store data`
11         **for** $j$ **to** $K$ **do**
12             sampl $\leftarrow$ get_cls_samples($\mathcal{D}, \text{cls}[j]$) ;   `// get samples for class cls[j]`
13             data$[j] \leftarrow$ random_select(sampl, $N + M$) ;   `// randomly select N+M samples`
14         $\mathcal{S}_i \leftarrow$ data$[:, :N]$ ;           `// first N samples`
15         $\mathcal{Q}_i \leftarrow$ data$[:, N :]$ ;        `// remaining M samples`
16         logits $\leftarrow$ model($\mathcal{S}_i, \mathcal{Q}_i$) ;        `// compute logits`
17         Initialize $y \leftarrow []$ ;            `// create labels`
18         **for** $j$ **to** $K$ **do**
19             $y[j] \leftarrow$ repeat($j, M$) ;       `// repeat class label j M times`
20         $y \leftarrow$ y.flatten() ;          `// flatten the label list`
21         loss $\leftarrow$ cross_entropy(logits, $y$) ;     `// compute loss`
22         acc$[i] \leftarrow$ calc_acc(logits, $y$) ;      `// compute accuracy`
       `// Do something with epoch results, e.g., store or print`

---

**Algorithm 2:** Inference Algorithm

---

**Input** : $N$ ;                           `// number of shots`
1   $K$ ;                             `// number of ways`
2   $M$ ;              `// number of query samples per class`
3   $\mathcal{D} \leftarrow \mathcal{D}_n$ ;              `// select the novel dataset`
4   $model \leftarrow$ load($model$) ;            `// load the best model`
5   num_iter ;                  `// number of iterations`
**Output:** Report novel accuracy (novel_acc)
6   Initialize novel_acc $\leftarrow []$ ;          `// list to store accuracies`
7   **for** $i$ **to** *num_iter* **do**
8      Create $\mathcal{S}_i, \mathcal{Q}_i, y$ ;          `// generate data and labels`
9      logits $\leftarrow$ model($\mathcal{S}_i, \mathcal{Q}_i$) ;         `// compute logits`
10      novel_acc$[i] \leftarrow$ calc_acc(logits, $y$) ;     `// compute accuracy`
11   novel_acc $\leftarrow$ mean(novel_acc) ;      `// calculate mean accuracy`
12   Report test accuracy (novel_acc)

---

### C.6   ADDITIONAL EXPERIMENTS ON INTERPRETATION OF ENSEMBLE FUSION

This section provides additional results on the interpretation of Ensemble Fusion reported in Figure 7a and Figure 7b.

Figure 10a compares the 5-shot novel class performance of DeepEMD, SimpleShot, Protonet, and FUSIONSHOT. We observe that FUSIONSHOT improves the top-5 performance of the best base model even though the worst model fails miserably. Furthermore, FUSIONSHOT can reduce failure rate in scenarios where the majority of the base models in an ensemble fail.

Figure 10b shows that a similar analysis we showed in Figure 7b is also applicable to the top-5 performance. Here we use SimpleShot as the latent distance function while changing the backbone architectures. The percentage of corrections made by the FUSIONSHOT ensemble is 20% when the minority of the base models in an ensemble make incorrect predictions and 10% when the majority

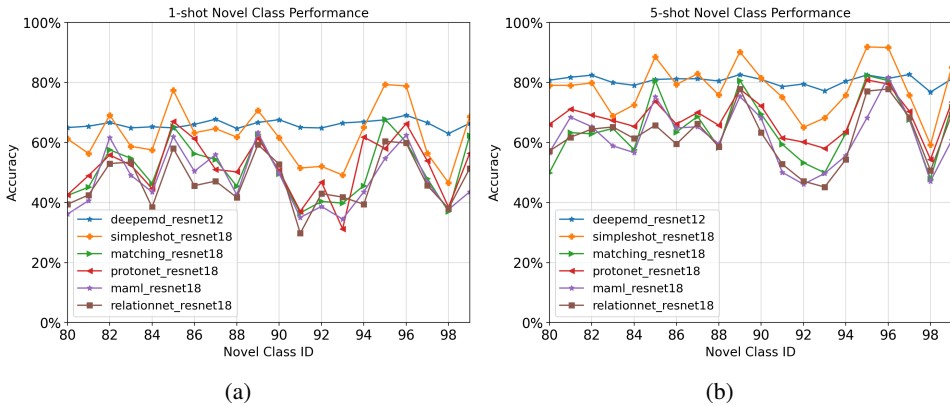

(a)  (b)

Figure 8: We show the performance of each base model on novel classes of *mini*-Imagenet, where each model is trained on *mini*-Imagenet for 1-shot-5way(a) and 5-shot-5way

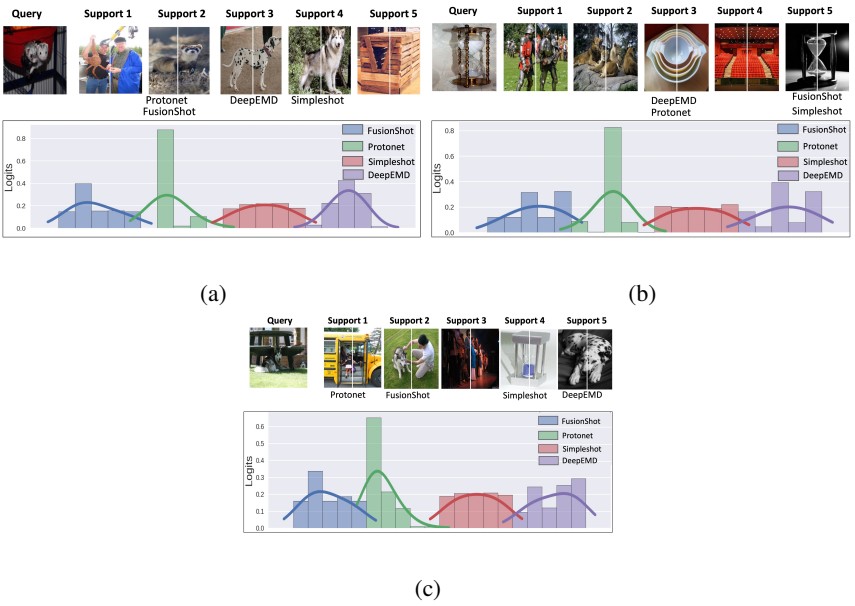

(a)  (b)

(c)

Figure 9: For given queries, we show the probability densities assigned by each model to the support images. We want to emphasize the importance of the secondary probabilities, i.e., the probabilities that do not correspond to the ground truth.

of the base models in an ensemble make incorrect predictions. Even when all of the base models in the ensemble make incorrect decisions, FUSIONSHOT selected ensembles can still improve the top-5 performance by correcting 1% of the errors.

## C.7 ADDITIONAL EXPERIMENTS FOR CROSS-DOMAIN EVALUATION

In addition to the results reported in Table 2, we perform FUSIONSHOT in the blind setting for a pool of base models without DeepEMD, which outperforms other base models by about 10%. Table 4 shows that FUSIONSHOT selected ensembles can improve the Simpleshot performance by 2% and improve other base model performances up to 6% when Simpleshot is removed.

## C.8 FUSIONSHOT VS PLURALITY VOTING

This set of experiments evaluates the performance gain of using MLP (learn-to-combine) in ensemble fusion stage. We compare FUSIONSHOT performance with plurality voting for all candidate

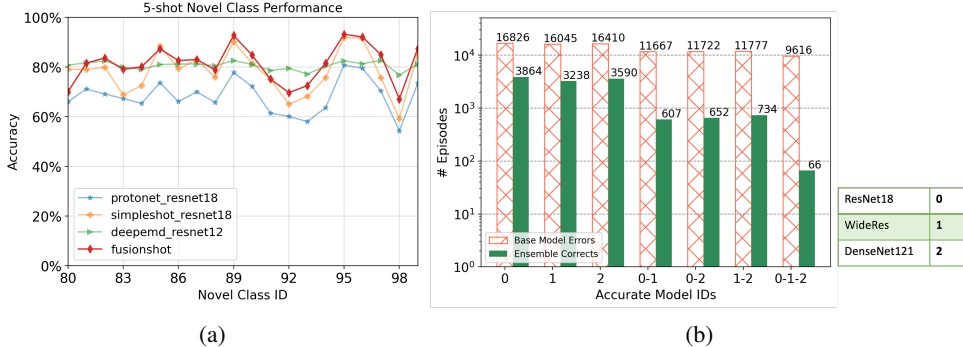

(a)            (b)

Figure 10: (a) We show the accuracy of base models and the FusionShot on each novel class out of 45000 episodes for the 5-shot 5-way performance. (b) We show the number of errors that each Simpleshot model made individually or together out of 45000 novel episodes for 1-shot 5-way settings in *mini*-Imagenet. The green bars show the number of corrected episodes by the ensemble model.

| Method | *mini*-Image→CUB (blind) |
|---|---|
| Matching | $52.17_{0.74}$ |
| Prototypical | $55.24_{0.72}$ |
| Relation | $50.93_{0.73}$ |
| MAML | $46.85_{0.72}$ |
| Simpleshot | $67.38_{0.70}$ |
| MAML-Matching-Proto | $59.82_{0.46}$ |
| MAML-Matching-Protonet-Relation | $61.17_{0.46}$ |
| MAML-Matching-Protonet-Relation-SimpleShot | $\mathbf{69.91}_{0.44}$ |

Table 4: Performing Cross-Domain experiments without DeepEMD. All the methods use the ResNet18 backbone.

ensemble sets (x-axis) by measuring the respective ensemble accuracy (y-axis). Figure 11 shows the sorted accuracy measurements for FUSIONSHOT learn to combine based ensemble fusion compared to FUSIONSHOT plurality voting consensus based ensemble fusion on the $\mathcal{Y}^{\text{novel}}$ dataset of *mini*-Imagenet benchmark.

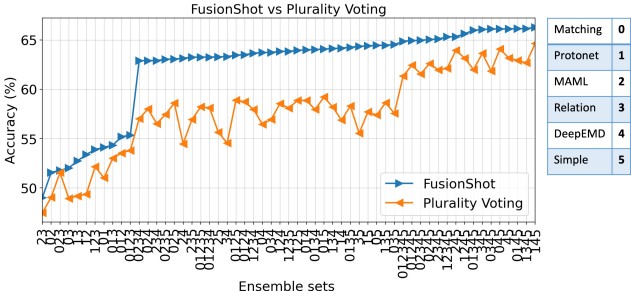

Figure 11: Performance of ensemble methods applied to the base predictions of Matching, Proto, MAML, Relation, DeepEMD, and SimpleShot methods with ResNet18 architectures.

## C.9 Additional Experiments on FusionShot Performance

### C.9.1 Backbone Feature Extractors and Latent Distance Method Analysis

Figure 12 provides some additional details for visual illustration of the results reported in Figure 6a and 6b. The hyperparameters, learning rate, number of epochs, and FusionShot architecture are kept the same. We use the same datasets, $\mathcal{Y}^{\text{train}}, \mathcal{Y}^{\text{val}}, \mathcal{Y}^{\text{novel}}$, during training, validation, and testing. We use the validation data to decide when to stop the learning process and select the best-performing validation model.

Figure 12a shows the performances of several FUSIONSHOT models, each is trained by using a different latent distance function under the same backbone DNN architectures. Figure 12b shows the performances of several FUSIONSHOT models, each is trained by using a different backbone DNN architecture but all using the same latent distance function for metric space comparison. We make two observations: (i) Our ensemble fusion approach outperforms the best-performing component base model, e.g., DeepEMD (64.21%). (ii) A fair number of ensembles can outperform DeepEMD (see those after the vetical line in both figures), indicating the opportunity of few-shot ensemble learning, and the opportunity for our focal diversity optimized ensemble pruning to effectively select top performing ensemble teams.

Table 5 and Table 6 shows all the ensemble teams included in Figure 12a and Figure 12b respectively using in a table, ranked by their novel accuracy in an ascending order. For Table 5, we make the following four highlights. (1) The best-performing 3-model ensemble set is highlighed by [1] in yellow. It shows the effectiveness of our focal diversity optimized ensemble pruning algorithm. (2) In contrast, by removing the Protonet base model and simply putting the two best-performing base models (DeepEMD and SimpleShot) into a 2-model ensemble does not yield the top-performing ensemble, see highlight by [2] in yellow. (3) The four-model ensemble highlighted by [3] in yellow cannot outperform the 2-model ensembles even when the two base models are selected from the component models of this 4-model ensemble. However, all of them are outperforming DeepEMD which has 64.21% novel accuracy. This shows that our ensemble fusion approach can effectively compose the top-performing ensmble learners that outperform the SOTA method. (4) As highlighted by [4] in yellow, we show that the best performing 4-model ensemble without including Simpleshot or DeepEMD as a member base model. This indicates the important role of a strong few-shot model in ensemble learning, as demonstrated by the top 16 performing ensemble teams identified by FUSIONSHOT, although we have shown that FUSIONSHOT can compose ensembles that outperform DeepEMD, as discussed in the observation (3).

Similar observations can be made for Table 6. (1) As highlighted by [1] in yellow, the best-performing ensemble is the 4-model team, each trained using different backbone architectures. (2) As highlighted by [2] in yellow, the 3-model ensemble has 66.70% novel accuracy lower than the top-performing 4-model ensemble (66.97%), but both outperform the ensemble with all 7 base models (66.13%). (3) We can also find the best performing 2-model ensemble that the majority of the models, highlighted by [3] in yellow, followed by the second best performing 2-model ensemble, highlighted by [4] in yellow, showing that the ensemble of the larger team size may not outperform the ensemble of smaller size.

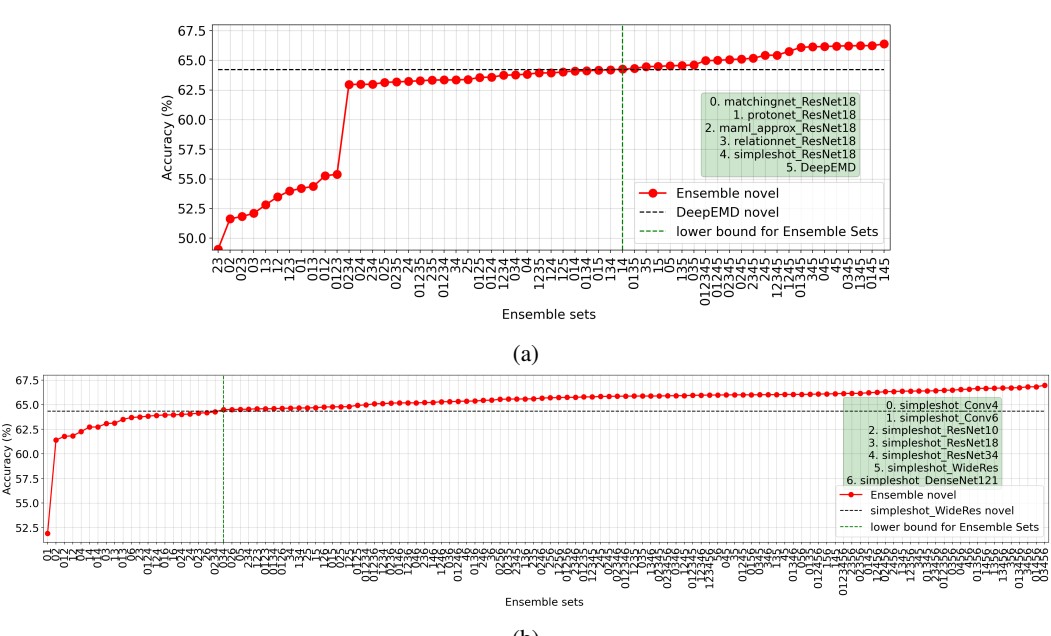

(a)

(b)

Figure 12: The sorted performances of each FusionShot model that is trained on each possible sub-ensemble set predictions in 1-shot 5-way setting on *mini*-Imagenet dataset. We provide all the scores in the Table 5 and 6 for more details.

| Ensemble Enumeration | Novel Accuracy | Ensemble Enumeration | Novel Accuracy |
|---|---|---|---|
| maml-rn | 49.06 | pn-maml-ss | 64.02 |
| mn-maml | 51.63 | mn-pn-EMD | 64.10 |
| mn-maml-rn | 51.82 | mn-pn-rn-EMD | 64.12 |
| mn-rn | 52.09 | mn-pn-ss | 64.17 |
| pn-rn | 52.81 | pn-rn-EMD | 64.20 |
| pn-maml | 53.48 | pn-EMD | 64.26 |
| pn-maml-rn | 53.98 | mn-pn-rn-ss | $64.31^{3}$ |
| mn-pn | 54.19 | rn-ss | 64.47 |
| mn-pn-rn | 54.37 | pn-ss | 64.48 |
| mn-pn-maml | 55.26 | mn-ss | 64.54 |
| mn-pn-maml-rn | $55.38^{4}$ | pn-rn-ss | 64.56 |
| mn-maml-rn-EMD | 62.96 | mn-rn-ss | 64.62 |
| mn-maml-EMD | 62.97 | mn-pn-maml-rn-EMD-ss | 64.97 |
| maml-rn-EMD | 62.98 | mn-pn-maml-EMD-ss | 65.00 |
| mn-maml-ss | 63.14 | mn-maml-rn-EMD-ss | 65.06 |
| mn-maml-rn-ss | 63.18 | mn-maml-EMD-ss | 65.11 |
| maml-EMD | 63.22 | maml-rn-EMD-ss | 65.18 |
| mn-pn-maml-rn-ss | 63.28 | maml-EMD-ss | 65.43 |
| maml-rn-ss | 63.34 | pn-maml-rn-EMD-ss | 65.43 |
| mn-pn-maml-rn-EMD | 63.34 | pn-maml-EMD-ss | 65.76 |
| rn-EMD | 63.35 | mn-pn-rn-EMD-ss | 66.10 |
| maml-ss | 63.39 | rn-EMD-ss | 66.14 |
| mn-pn-maml-ss | 63.56 | mn-EMD-ss | 66.16 |
| mn-pn-maml-EMD | 63.58 | EMD-ss | $66.20^{2}$ |
| pn-maml-rn-EMD | 63.75 | mn-rn-EMD-ss | 66.22 |
| mn-rn-EMD | 63.77 | pn-rn-EMD-ss | 66.23 |
| mn-EMD | 63.83 | mn-pn-EMD-ss | 66.24 |
| pn-maml-rn-ss | 63.94 | pn-EMD-ss | $66.38^{1}$ |
| pn-maml-EMD | 63.94 | | |

Table 5: Comparison of FusionShot performance for each Few-shot Learning method combination for ResNet18 in 1-shot 5-way setting on *mini*-Imagenet dataset. The highlighted data descriptions are in Appendix Section D.4.1

| Ensemble Enumeration | Novel Accuracy | Ensemble Enumeration | Novel Accuracy |
|---|---|---|---|
| C4-C6 | 51.91 | RN10-WR-DN121 | 65.70 |
| C4-RN10 | 61.40 | C6-RN10-WR-DN121 | 65.72 |
| C4-C6-RN10 | 61.78 | C4-C6-RN10-WR-DN121 | 65.74 |
| C6-RN10 | 61.83 | RN10-RN18-RN34-DN121 | 65.74 |
| C4-RN34 | 62.26 | C4-C6-RN10-RN18-WR | 65.78 |
| C6-RN34 | 62.73 | C6-RN10-RN18-RN34-WR | 65.79 |
| C4-C6-RN34 | 62.73 | RN10-RN34-WR | 65.84 |
| C4-RN18 | 63.09 | C4-RN10-RN34-WR | 65.84 |
| C6-RN18 | 63.11 | C4-RN10-RN18-RN34-DN121 | 65.86 |
| C4-C6-RN18 | 63.50 | C4-C6-RN10-RN18-RN34-DN121 | 65.86 |
| C4-DN121 | 63.69 | C6-RN10-RN18-WR | 65.87 |
| RN10-RN18 | 63.75 | C4-RN18-WR | 65.87 |
| C4-C6-RN10-RN34 | 63.83 | C6-RN18-RN34-DN121 | 65.88 |
| C6-RN10-RN34 | 63.90 | C4-RN10-RN18-RN34-WR | 65.88 |
| C4-C6-DN121 | 63.95 | C4-RN10-RN18-RN34-WR-DN121 | 65.90 |
| C6-DN121 | 63.96 | C4-RN18-RN34-DN121 | 65.91 |
| C4-RN10-RN34 | 64.01 | C6-RN10-RN34-WR | 65.91 |
| RN10-RN34 | 64.05 | C4-C6-RN10-RN18-RN34-WR | 65.95 |
| C4-RN10-RN18 | 64.13 | C6-RN10-RN18-RN34-DN121 | 65.96 |
| RN10-DN121 | 64.18 | C6-RN10-RN18-RN34-WR-DN121 | 65.98 |
| C4-RN10-RN18-RN34 | 64.27 | WR-DN121 | 65.98 |
| C4-RN18-RN34 | 64.50 | C4-RN34-WR | 65.99 |
| C4-RN10-DN121 | 64.50 | RN18-WR | 65.99 |
| C4-WR | 64.52 | C4-C6-RN10-RN34-WR | 66.00 |
| RN10-RN18-RN34 | 64.54 | C4-C6-WR-DN121 | 66.00 |
| C6-RN10-RN18 | 64.58 | C4-RN18-RN34-WR | 66.01 |
| C4-C6-RN10-RN18 | 64.58 | RN18-RN34-DN121 | 66.02 |
| C4-C6-RN18-RN34 | 64.60 | C6-RN18-WR | 66.02 |
| C4-C6-RN10-DN121 | 64.63 | RN34-WR | 66.04[3] |
| RN18-RN34 | 64.64 | C4-C6-RN18-RN34-DN121 | 66.04 |
| C6-RN18-RN34 | 64.66 | C4-WR-DN121 | 66.04 |
| RN10-WR | 64.67 | C4-C6-RN18-WR | 66.05 |
| C6-WR | 64.69 | C4-C6-RN10-RN34-WR-DN121 | 66.08 |
| C6-RN10-DN121 | 64.76 | C6-WR-DN121 | 66.09 |
| C4-C6-WR | 64.78 | C6-RN34-WR | 66.11 |
| C4-RN10-WR | 64.78 | C4-C6-RN10-RN18-RN34-WR-DN121 | 66.13 |
| C6-RN10-WR | 64.80 | RN10-RN18-WR-DN121 | 66.14 |
| C4-C6-RN10-WR | 64.94 | C4-RN10-RN18-WR-DN121 | 66.14 |
| C4-C6-RN10-RN18-RN34 | 64.97 | C4-C6-RN34-WR | 66.22 |
| C4-C6-RN10-RN18-DN121 | 65.09 | C6-RN10-RN34-WR-DN121 | 66.25 |
| C6-RN10-RN18-RN34 | 65.10 | C4-RN10-RN34-WR-DN121 | 66.32 |
| C4-RN10-RN18-DN121 | 65.16 | RN10-RN34-WR-DN121 | 66.33 |
| C4-C6-RN34-DN121 | 65.17 | C6-RN18-RN34-WR | 66.37 |
| C6-RN10-RN18-DN121 | 65.18 | C6-RN10-RN18-WR-DN121 | 66.37 |
| C4-RN34-DN121 | 65.18 | RN18-RN34-WR | 66.39 |
| RN10-RN18-DN121 | 65.21 | C4-C6-RN18-RN34-WR | 66.39 |
| C6-RN34-DN121 | 65.22 | RN10-RN18-RN34-WR-DN121 | 66.42 |
| C6-RN10-RN34-DN121 | 65.30 | C4-C6-RN10-RN18-WR-DN121 | 66.47 |
| C4-RN18-DN121 | 65.30 | C4-RN18-WR-DN121 | 66.47 |
| C4-C6-RN10-RN34-DN121 | 65.33 | C4-RN34-WR-DN121 | 66.54 |
| RN34-DN121 | 65.36[4] | RN34-WR-DN121 | 66.58 |
| C4-C6-RN18-DN121 | 65.37 | C4-C6-RN18-WR-DN121 | 66.65 |
| RN10-RN34-DN121 | 65.44 | C6-RN34-WR-DN121 | 66.66 |
| RN18-DN121 | 65.45 | C6-RN18-WR-DN121 | 66.67 |
| C4-RN10-WR-DN121 | 65.54 | C6-RN18-RN34-WR-DN121 | 66.69 |
| C4-RN10-RN18-WR | 65.57 | RN18-WR-DN121 | 66.70[2] |
| RN10-RN18-RN34-WR | 65.57 | C4-C6-RN18-RN34-WR-DN121 | 66.73 |
| C6-RN18-DN121 | 65.58 | RN18-RN34-WR-DN121 | 66.81 |
| RN10-RN18-WR | 65.59 | C4-C6-RN34-WR-DN121 | 66.82 |
| C4-RN10-RN34-DN121 | 65.67 | C4-RN18-RN34-WR-DN121 | 66.97[1] |

Table 6: Comparison of FusionShot performance for each Few-shot Learning architecture combination for SimpleShot in 1-shot 5-way setting on *mini*-Imagenet dataset. The highlighted data descriptions are in Appendix Section D.4.1

# D   MORE DISCUSSIONS ON FEW-SHOT RECENT LITERATURE

Few-shot image classifiers can be categorized based on their learning strategies into two types: inductive and transductive. The transductive learning methods can provide high accuracy, (Liu et al., 2018), but they require comparison with the fitted data for every new query which limits their applicability to other datasets, and slows the inference time performance. The inductive methods, however, aim for a task-agnostic function that can be applied to any dataset. In this work, we focus on the few-shot classifiers that follow inductive learning.

Numerous efforts have been dedicated to improving the performance of the few shot classifiers in the inductive setting by either designing a more complex latent distance function or improving the expressiveness of the feature extractor. The precedential methods, (Koch et al., 2015; Snell et al., 2017; Vinyals et al., 2016) focused on the non-parametric latent distance functions such as Manhattan, Euclidian, and cosine distances. Following these works, relation networks (Sung et al., 2018), and recent Graph-CNN variants (Garcia & Bruna, 2017; Kim et al., 2019) offered the parameterized approaches by learning the distance function with neural networks. The SOTA works show the importance of the correlations between the embeddings by measuring the joint distribution with parameterized metrics, such as Earth Mover's Distance (Zhang et al., 2020), Mahalanibis distance (Bateni et al., 2020), and Brownian Distance (Xie et al., 2022).

On the other side, various works that focus on the feature extractor proposed different techniques for training and testing. (Chen et al., 2019) performed a supervised pre-training on the feature extractor fraught with a fully connected layer that is fine-tuned during test time with few samples. Subsequently, (Wang et al., 2019) simply implemented k-nearest-neighbor to the outputs of a pre-trained feature extractor and showed the performance with various feature extractor architectures. (Tian et al., 2020) defend that pre-trained embedding can outperform many classical methods. To further enhance the feature extractor's expressiveness, several approaches have been proposed, including the addition of extra self-supervised loss (Mangla et al., 2020), the utilization of augmentation techniques (Luo et al., 2021; Yang et al., 2021), and the generation of supplementary data (Li et al., 2020). Research on feature extractor adaptation during test time (Finn et al., 2017; Rusu et al., 2018) finds the best initial parameters for learning novel tasks in testing, and (Ye et al., 2020; Bateni et al., 2020) shows adaptive embedding to the target class during testing.

In terms of task-agnostic and generalization capability of few-shot learners, the literature interest shifted towards improving the generalization of few-shot learners in unseen datasets (Triantafillou et al., 2019), which mirrors the setting we experimented with in the blind cross-domain setting. The focus is on creating a universal representation to generalize to unseen domains (Bronskill et al., 2020; Liu et al., 2020; Triantafillou et al., 2021), and creating few-shot learners that are able to distinguish common features in between datasets (Dvornik et al., 2020; Li et al., 2021). A recent approach, (Hiller et al., 2022) proposes the use of Vision Transformers to overcome the lack of fine-grained labels and learn high-level statistics.

Many methods, including universal representation learners (Li et al., 2021), have enhanced the generalization of their approaches by employing multiple backbones. (Dvornik et al., 2019) used ensemble methods to mitigate the variance of few-shot learning classifiers, and jointly train their multiple backbones during supervised training with different penalizing terms for diversity and cooperation. Similarly, (Bendou et al., 2022) exploited the multiple backbones and improved the expressiveness of each backbone with self-supervised loss. In terms of leveraging other methods, (Ma et al., 2021) leveraged dual prototype networks, while one model is pre-trained to regularize the learning of the main encoder. Most recently, (Ma et al., 2022) proposed a geometric ensemble approach by using Voronoi Diagrams to model class relationships on the latent space.

