# OpenReview forum: "FusionShot: Boosting Few Shot Learners with Focal-Diversity Optimized Ensemble Method"
_ICLR.cc/2024/Conference — ICLR 2024 Conference Withdrawn Submission_

### Official Review · Reviewer_K6te · 2023-10-30

**Soundness:** 2 fair
**Presentation:** 2 fair
**Contribution:** 2 fair
**Rating:** 3
**Confidence:** 5

**Summary:**

This paper proposes to use an ensemble method to improve Few Shot Learners. Ensemble of backbones generation is proposed by either using various  latent distance methods on a given backbone or by providing backbones with various architecture. Given a set of backbones, ensemble fusion is performed using a MLP network on the classification  probabilities outputs $\{\hat{y]_i} from the various backbones.
Additionally pruning of the ensemble of backbones is proposed based on Focal Diversity. Focal Diversity proposes a metric based on the diversity encountered with respect to other models's predictions when a model fails. Then the best pruned ensemble of models is selected based on a combination of focal diversity and accuracy measured on training dataset. To reduce the complexity related to the search of this best pruned ensemble, a genetic algorithm is proposed. The speeding gain being more evident when the number of base models increase.
Experimental experiments evaluates the benefits of the proposed approach. First the impact of the generation of the ensemble of backbones (latent distance ensemble, feature extraction ensemble) is studied. Then the impact of selection of the best pruning ensemble is reported.

**Strengths:**

Using ensemble methodology for Few Shot Learning sounds relevant. Experimental results show improved performance in the context where it was done. Especilly using a pruned ensemble is interesting from the study reporting variations related to the pruned ensemble retained.

The use of genetic algorithm to speed up the search of pruned ensemble is quite efficient.

**Weaknesses:**

Overall, the ensemble fusion technique proposed is sound and valuable. However there are many points that need clarifications. First baselines considered in experiments are rather weak compared to current state of the art. Second no comparison is provided with respect to other proposed ensemble method such as the one proposed by Bendou et al 2022. Finally there are some points that needs to be clarified (e.g. what type of MLP is used for fusion, what is the performance of the pruning algorithm).

In abstract it is mentioned in the set of contributions: 'First we revisit he FSL architectures to analyse why some FSL perform well whereas other [..]'. This is not provided in the paper.

The overall organisation of the paper could be improved. For instance section 2 is quite dense and could be significantly reduced. This would let room to get more technical insights (there is only one equation referenced in main paper) and to have more information on overall performance

The fusion of ensemble predictions is quickly presented via figure 3 and with equation 4 in appendix. This fusion should be more explicit in the main paper. First why $y$ outputs from backbones are probabilities and not usual outputs (log-prob) ? (in main paper, it is mentioned probabilities while in appendix prior to Equation 4, it is mentioned logits...) Then it is not clear what are the properties of the MLP. From Equation 4, we could think of a fully connected layer.... but then this would mean that input probability of class $c_1$ could influence output probability of class $c_2$.... but since classes will change from one episode to another, this could be not consistent. So I guess that shape of $W_{\gamma}$ is constrained to be just a kind of weighted sum per model, but this is not explicited. This needs to be clarify. Also not that $i$ index is either used for class or for model. This is quite confusing.

The ensemble pruning is based on focal diversity. Presentation of this method could be improved (some details could be put in appendix). Especially justification of criterion used could be improved. There are some mixing between $m$ and $M$. To select best pruned ensemble a weighted metric combining focal diversity and accuracy is proposed. It is not mentioned what are the typical values for $w_1$ and $w_2$. Also an additional $\lambda$ parameter is introduced in appendix while not being discussed about in main paper. Is it used? Also on which dataset is performed the measurement? is it on the train dataset or the validation dataset? What impact of choosing one or another? Finally although an automatic pruning selection is presented, there is no evidence of the performance of this selection process. Discussions only deal with the best pruned version... that optimal pruning may not be the optimal one as observed for training on Fig 4.a.

When considering performance results reported in table 1 and 2, these results are quite below state of the art. For instance if we consider DeepEMD,  performance reported in table 2 is 64.21 & 80.51, while DeepEMD paper reports up to 68.77 & 84.13 performance. Furthermore no comparison is made with other ensemble approach such as the one proposed in Bendou and al 2022. Bendou and all reports significantly higher performance. Also there is an underlying consideration to consider when using ensemble approach. Using several backbone increase significantly the performance of the proposed approach. Thus one could consider if it is worth increasing performance via such ensemble approach or using a higher complexity model. Note that in Bendo et al, comparison are proposed at same complexity order.
Also considering only two datasets for evaluation is quite weak here. Additional experiments with other datasets should be considered.

**Questions:**

1. What about the additional complexity of using ensemble methods?
2. How does this method compare with respect of work of Bendou et al 2022? (in terms of perfomance and complexity)
3. What is effective efficiency from the proposed pruning strategy? (what about performance of selected pruned ensemble versus best pruned ensemble?)
4. Why using output probabilities from model rather than log-probabalities?
5. What is the benefit of using MLP fusion rather than simple averaging? (especially if considering that pruning is considered to remove unwanted models from the input ensemble)
6. How does the proposed method behaves when considering stronger baseline.

---

> ### Author Response · Authors · 2023-11-17
> **Thanking Reviewer's Comments**
>
> We would like to thank the Reviewer for these valuable comments and appreciate the effort and time spent on our paper. We will address the listed concerns. However, due to the time constraint, we decided to withdraw the paper.
>
> Thank you.

---

### Official Review · Reviewer_Mcgk · 2023-10-31

**Soundness:** 3 good
**Presentation:** 3 good
**Contribution:** 2 fair
**Rating:** 5
**Confidence:** 4

**Summary:**

The authors propose the method of "FusionShot" where they explore the methodology of ensemble-based approaches applied to few-shot learners. Specifically, they delve into two aspects of model fusion which are distance metrics and embedding spaces learned by DNNs. They set up a pool of base few-shot learners and train their FusionShot model to choose the best performing learners for a given problem. To validate the effectiveness of their approach, the authors perform extensive experimental evaluations under various settings.

**Strengths:**

- The motivation for the proposed method is clear and straightforward, with simple formulation that is easy to understand.

- Their methodology is technically plausible, while solving a practical problem with possibility for application to wide range of few-shot classification tasks.

- Their approach shows competitive performance on some well-known few-shot learning and cross-domain few-shot learning benchmarks.

- They validate their method under various base learners and hyperparameter settings with extensive ablation experiments to back up their claims.

**Weaknesses:**

- Although extensive ablation experiments on miniImageNet are well-appreciated and helpful for understanding the characteristics of the proposed method, there are other well-known, more general few-shot learning benchmarks such as MetaDataset [a] and tieredImageNet. Additional validations will further help to back up the effectiveness of the proposed method.

  [a] Meta-Dataset: A Dataset of Datasets for Learning to Learn from Few Examples, Triantafillou et al., ICLR 2020

- Contrary to the authors' claim of solving general few-shot learning problems, the proposed method seems viable only for few-shot "classification" methods, where metric space is constructed with discriminative training/loss and proposed focal diversity metric can be measured. Since model-agnostic, gradient-based methods such as MAML, Reptile, and CAML can be used to solve few-shot regression problems, can this method be extended to solve both problems?

- By looking at Table 1, improvements from fusion channel of backbones seems greater than fusion channel of distances. Since the main contribution is to utilize various embedding and metric spaces of various few-shot classification algorithms, these results seems counterintuitive. Are there any further analysis on these results?

- Minor point: The contributions described in the abstract and the introduction section seems somewhat vague, providing little details about the proposed methodology. I would suggest changing some sentences to provide a more accurate and concrete description of the proposed "fusion training" based on "focal diversity metrics". Also, the notations seem to be overly cluttered and redundant without clear purpose.

**Questions:**

Please refer to the weaknesses above. I appreciate the insights and motivation provided in the paper, but there are some limitations in the method's generalizability and its experimental validations.

---

> ### Author Response · Authors · 2023-11-17
> **Thanking Reviewer's Comments**
>
> We would like to thank the Reviewer for these valuable comments and appreciate the effort and time spent on our paper. We will address the listed concerns. However, due to the time constraint, we decided to withdraw the paper.
>
> Thank you.

---

### Official Review · Reviewer_mohu · 2023-11-02

**Soundness:** 2 fair
**Presentation:** 2 fair
**Contribution:** 2 fair
**Rating:** 3
**Confidence:** 4

**Summary:**

The paper tackles the problem of few-shot image classification using ensemble methods. The authors propose a new method to ensemble predictions from different independent few-shot learners. The diversity of the few-shot learners comes either from using different networks as the backbone, or using different distance metrics in the classifier. The proposed emsembling strategy includes two key components: an MLP that is trained to fuse the ensemble predictions, and a selection policy for the ensemble members, based on the diversity and cooperation among the ensemble members.

**Strengths:**

- Technical parts of the paper are clear and well-explained
- The idea of picking only a subset of the available few-shot learners to construct a better overall ensemble is interesting. Specifically, Figure 4a is very representative.
- The method shows competitive with SoTA results.

**Weaknesses:**

- I have some questions to the claims made in the introduction (without any references) that are unclear or seem wrong:
    - “However, in image classification, designing an optimal few-shot learner that can persistently outperform the state-of-the-art (SOTA) methods seems stalled in recent years for a number of reasons.” Is that so? What can we say about zero- or few-shot performance of CLIP [A] and similar models? It seems like the field is moving forward with large-scale Image-Text models.
    - “First, it is extremely hard to train an individual few-shot learner that can compete and win persistently” What do the authors mean exactly by “compete and win persistently”?
    - “Second, unlike conventional deep neural networks (DNNs), e.g., auto encoder-decoder and CNN, which learn invariant and distinctive features about data by building high-level features from low-level ones through learning hierarchical feature representations; the few-shot learner employs the metric space learning, a distance metric based approach, to learn deep embedding of input data for latent feature extraction.” This seems wrong as one does not contradict the other. Few-shot learners can be CNNs and auto-encoders, and the main reason they work for image classification is exactly because they rely on hierarchical representations and encode invariances in their weights.
    - “By combining deep embedding learning with metric space distance learning through metric-space loss optimization, the few-shot learners can map examples of similar features in the real world (input data space) to the latent neural embedding representations with dual properties: (i) the examples of similar features in the real world will be mapped to the latent feature vectors that are closer in the latent embedding space, and (ii) the examples with large dissimilarity in the real world will be sharply distant in the latent feature embedding space” Again, this is not an exclusive property of few-shot learners. It has been long discovered that many vision neural networks, even the ones not specifically trained to do so, map similar visual concepts to similar vectors, see [A, B] for example. Please note that there are ways to perform few-shot image classification with methods that were not trained to do so explicitly, and they still work well, see [C]
- The method doesn’t seem as novel as it is presented. The claim in the introduction “FUSIONSHOT is the first to ensemble multiple independently trained few-shot models through integrating learning to combine and focal diversity optimization” is not correct. The paper of [D] proposes to take independent feature extractors and aggregate their features at test time for few-shot learning.
- From what I understand, to train the learnable fusion of the ensembles (implemented as an MLP), the method assumes that the ensemble members must be fixed apriori, and the corresponding feature dataset must be generated. This is inefficient both in terms of training and inference. For example, if we were to experiment with a different set of few-shot learners in our ensemble, we would need to generate their features (if not yet generated) and re-train the fusion mechanism. Also, if at test time we have a new few-shot learner available for ensembling, we will not be able to use it since the fusion was not trained on it.
- The ensemble pruning strategy seems a little ad-hoc.

[A] Radford, et al. "Learning transferable visual models from natural language supervision”, 2021
[B] Caron, et al. "Emerging Properties in Self-Supervised Vision Transformers”, 2021
[C] Dvornik, et al. “"Diversity with cooperation: Ensemble methods for few-shot classification.”, 2019
[D] Dvornik, et al. "Selecting Relevant Features from a Multi-domain Representation for Few-shot Classification”, 2020

**Questions:**

When is exactly selection of the ensemble members performed, before or after training the fusion MLP? I suspect it has to be done before training the MLP, but I could not find it in the text.

**Details Of Ethics Concerns:**

-

---

> ### Author Response · Authors · 2023-11-17
> **Thanking Reviewer's Comments**
>
> We would like to thank the Reviewer for these valuable comments and appreciate the effort and time spent on our paper. We will address the listed concerns. However, due to the time constraint, we decided to withdraw the paper.
>
> Thank you.